# A folate inhibitor exploits metabolic differences in *Pseudomonas aeruginosa* for narrow-spectrum targeting

Connor Chain[1], Joseph P. Sheehan[1], Xincheng Xu[2,3], Soodabeh Ghaffari[1], Aneesh Godbole[4], Hahn Kim[3,5], Joel S. Freundlich[4,6], Joshua D. Rabinowitz[2,3,7] & Zemer Gitai[1] ✉

*Pseudomonas aeruginosa* is a leading cause of hospital-acquired infections for which the development of antibiotics is urgently needed. Unlike most enteric bacteria, *P. aeruginosa* lacks enzymes required to scavenge exogenous thymine. An appealing strategy to selectively target *P. aeruginosa* is to disrupt thymidine synthesis while providing exogenous thymine. However, known antibiotics that perturb thymidine synthesis are largely inactive against *P. aeruginosa*.

Here we characterize fluorofolin, a dihydrofolate reductase (DHFR) inhibitor derived from Irresistin-16, that exhibits significant activity against *P. aeruginosa* in culture and in a mouse thigh infection model. Fluorofolin is active against a wide range of clinical *P. aeruginosa* isolates resistant to known antibiotics. Metabolomics and in vitro assays using purified folA confirm that fluorofolin inhibits *P. aeruginosa* DHFR. Importantly, in the presence of thymine supplementation, fluorofolin activity is selective for *P. aeruginosa*. Resistance to fluorofolin can emerge through overexpression of the efflux pumps MexCD-OprJ and MexEF-OprN, but these mutants also decrease pathogenesis. Our findings demonstrate how understanding species-specific genetic differences can enable selective targeting of important pathogens while revealing trade-offs between resistance and pathogenesis.

The discovery of penicillin ushered in a golden era of antibiotic discovery, but the lack of discovery of antibiotics, coupled with a rise in resistance to known antibiotics, is producing a growing public health crisis[1–3]. The Gram-negative opportunistic pathogen, *Pseudomonas aeruginosa*, is of particular interest for antibiotic development as it has evolved multiple mechanisms to evade antibiotics including a robust outer membrane[4,5], multiple efflux pumps[6–8] and other antibiotic resistance determinants such as carbapenamases[9]. *P. aeruginosa* is often associated with chronic infections and the resulting prolonged exposure to antibiotics can have detrimental health effects due to microbiome

[1]Department of Molecular Biology, Princeton University, Princeton, NJ, USA. [2]Lewis-Sigler Institute for Integrative Genomics, Princeton University, Princeton, NJ, USA. [3]Department of Chemistry, Princeton University, Princeton, NJ, USA. [4]Department of Pharmacology, Physiology and Neuroscience, Rutgers University - New Jersey Medical School, Newark, NJ, USA. [5]Small Molecule Screening Center, Princeton University, Princeton, NJ, USA. [6]Division of Infectious Disease, Department of Medicine and the Ruy V. Lourenço Center for the Study of Emerging and Re-emerging Pathogens, Rutgers University - New Jersey Medical School, Newark, NJ, USA. [7]Ludwig Institute for Cancer Research, Princeton Branch, Princeton University, Princeton, NJ, USA. ✉e-mail: zgitai@princeton.edu

disruption[10–12]. Most antibiotics do not exhibit substantial efficacy against *P. aeruginosa* and there are no commercial narrow-spectrum antibiotics that selectively target *P. aeruginosa*[13].

There is mounting evidence for the potential benefits of antibiotics with narrow species selectivity, as they can target pathogens of interest with minimal disruption to the host microbiome[14,15]. Existing strategies for narrow-spectrum targeting have largely focused on developing drugs whose targets are only present in specific pathogens[16–20]. However, there are also species-specific genetic and metabolic differences that could hypothetically be exploited to cause antibiotics with widely conserved molecular targets to only kill specific species of interest. In particular, unlike most enteric bacteria, *P. aeruginosa* lacks thymidine kinase and thymidine phosphorylase activity, and is therefore unable to scavenge thymine from the environment[21–23] (Fig. 1a). We therefore hypothesized that targeting de novo thymidylate synthesis with concurrent thymine supplementation could selectively target *P. aeruginosa*.

Dihydrofolate reductase (DHFR) catalyses the production of tetrahydrofolate, which is required for de novo thymidylate synthesis[24]. There are known DHFR inhibitors, such as the commercially used antibiotic trimethoprim (TMP)[25] and Irresistin-16 (IRS-16)[26], a compound that can simultaneously disrupt membrane integrity and inhibit DHFR activity. However, *P. aeruginosa* is intrinsically resistant to all known DHFR inhibitors, including both TMP and IRS-16 (refs. [26,27]). Here we characterize a DHFR inhibitor, fluorofolin, that shows potent activity against *P. aeruginosa* in vitro and in a mouse model. We use fluorofolin to both selectively eliminate *P. aeruginosa* from mixed-species bacterial cultures and to uncover a trade-off between *P. aeruginosa* antibiotic resistance and virulence. These studies demonstrate the utility of inhibitors of broadly conserved targets as both narrow-spectrum agents and tools for discovery of context-specific features of pathogenesis.

## Fluorofolin inhibits DHFR in *P. aeruginosa*

*P. aeruginosa* is resistant to known DHFR inhibitors, such as TMP (minimum inhibitory concentration (MIC) > 100 µg ml$^{-1}$) and IRS-16 (MIC > solubility limit of ~50 µg ml$^{-1}$). However, while characterizing fluorofolin, a derivative of IRS-16, we discovered that it exhibited potent activity against *P. aeruginosa* PA14 (MIC = 3.1 µg ml$^{-1}$). Fluorofolin has broad-spectrum antibiotic activity, as it was also capable of inhibiting growth of two other strains of *P. aeruginosa*, PA01 and ATTC 27853, as well as all 5 of the ESKAPE[28] pathogens tested at concentrations less than 50 µg ml$^{-1}$ (Table 1).

Fluorofolin is a pyrroloquinazolinediamine derivative that is functionalized with a 2-fluoropyridine (Fig. 1b). Unlike IRS-16, which is bactericidal[26], fluorofolin exhibited bacteriostatic activity in rich media (Fig. 1c). To characterize its mechanism of action, we examined the ability of fluorofolin to directly inhibit the enzymatic activity of purified *E. coli* DHFR (FolA). Fluorofolin inhibited DHFR activity to half its maximum value (IC$_{50}$) at a concentration of 2.5 ± 1.1 nM, which was comparable to that of TMP (IC$_{50}$ of 8.7 ± 3.6 nM) (Fig. 1d). Fluorofolin also exhibited modest specificity for bacterial DHFR in vitro; in an analogous assay using purified human DHFR, fluorofolin had an IC$_{50}$ of 14.0 ± 4 nM (Fig. 1e).

To test the ability of fluorofolin to inhibit *P. aeruginosa* DHFR in vivo, we performed metabolomics. As a positive control for DHFR inhibition, we turned to TMP. *P. aeruginosa* is resistant to TMP at the highest doses tolerated by humans and is thus not clinically useful in targeting *P. aeruginosa* infections, but we found that TMP can inhibit the growth of *P. aeruginosa* cultures at very high concentrations in vitro (PA14 MIC = 125 µg ml$^{-1}$, ~70 times higher than the mean steady-state serum concentration of TMP achievable in serum after clinical use of cotrimoxazole[29]). Metabolomics demonstrated that both TMP and fluorofolin treatment caused significant upregulation of the purine and dTTP intermediates 5-aminoimidazole-4-carboxamide ribonucleotide (AICAR) and 2′-deoxyuridine 5′-monophosphate (dUMP) in

*P. aeruginosa* ($P$ = 0.0186 and 0.0196, respectively) (Fig. 1f). These results support the conclusion that fluorofolin inhibits folate metabolism in *P. aeruginosa* cells.

To predict how fluorofolin might interact with *P. aeruginosa* DHFR, we performed molecular docking. As the structure of *P. aeruginosa* DHFR has not been solved, we used AlphaFold[30,31] to generate a homology model for its three-dimensional structure. We inserted the NADPH cofactor into this structure and aligned it to the experimentally derived structure of NADPH and TMP bound to *E. coli* DHFR[32] (Extended Data Fig. 1a). We then used AutoDock Vina[33,34] to predict the binding of *P. aeruginosa* DHFR to either fluorofolin (Extended Data Fig. 1b) or TMP (Extended Data Fig. 1c). Both fluorofolin and TMP were predicted to bind DHFR in the dihydrofolate binding pocket and fluorofolin was predicted to have a lower binding affinity (−9.104 kcal mol$^{-1}$) than TMP (−6.8 kcal mol$^{-1}$). The stronger binding of fluorofolin may be explained by an additional hydrogen bond between Leu23 of *P. aeruginosa* DHFR and the pyridine group of fluorofolin (Extended Data Fig. 1d) that is not formed with TMP.

As IRS-16 was shown to kill bacteria through both the inhibition of DHFR and disruption of bacterial membranes[26], we wanted to test the ability of fluorofolin to permeabilize *P. aeruginosa* PA14. We observed that fluorofolin does not cause significant disruption of *P. aeruginosa* PA14 membrane permeability (Extended Data Fig. 2a). We also confirmed that fluorofolin does not cause membrane depolarization or permeabilization in *E. coli lptD4213* (ref. [35]) (the membrane polarization reporter DiOC2(3) does not work in *P. aeruginosa* (Extended Data Fig. 2b)). We hypothesized that this lack of membrane targeting might also decrease toxicity. As predicted, fluorofolin did not cause significant permeabilization of red blood cell membranes at biologically relevant concentrations (Extended Data Fig. 2c) and displayed less toxicity than IRS-16 towards several mammalian cell types, including a 9.6-fold increase in IC$_{50}$ against PBMCs and a 58.5-fold increase in IC$_{50}$ against WI-38 cells compared with IRS-16 (Extended Data Fig. 2d).

As fluorofolin and TMP exhibit similar functional inhibition of purified DHFR, we wanted to investigate why fluorofolin was better at inhibiting *P. aeruginosa* growth. We hypothesized that fluorofolin may better accumulate inside of *P. aeruginosa*. *P. aeruginosa* is particularly drug resistant due to its robust outer membrane and expression of multiple RND-type efflux pumps[36]. Using mass spectrometry to measure drug accumulation[37,38], we found that fluorofolin accumulated in *P. aeruginosa* more rapidly (Fig. 1g) and to higher levels (Fig. 1h) than TMP. It was previously shown that the constitutively active efflux pump MexAB-OprM can export TMP[27]. We confirmed that PA14 mutants with disruptions in *mexA*, *mexB* and *oprM* showed large reductions in MIC to TMP. However, these mutants did not decrease their MIC against fluorofolin to the same extent (Extended Data Table 1), suggesting that the increased accumulation of fluorofolin is in part due to decreased efflux of fluorofolin by MexAB-OprM. We also note TMP still had a higher MIC than fluorofolin in *mexA*, *mexB* and *oprM* mutants, suggesting that fluorofolin may not be a good substrate for other efflux pumps or may improve other features of cell accumulation such as membrane penetrance.

## Fluorofolin inhibits murine thigh infection of *P. aeruginosa*

We next sought to determine whether fluorofolin has activity in an in vivo mouse infection model. In mice, fluorofolin displayed favourable plasma protein binding (71.7% bound, 91.9% recovery, Extended Data Table 2). Upon oral administration, fluorofolin achieved a peak concentration of 4.0 µg ml$^{-1}$ with a half-life of 12.1 h (Fig. 2a). Because the peak plasma concentration was so near the MIC for PA14 (3.1 µg ml$^{-1}$), we sought to further potentiate fluorofolin's antibiotic activity. Sulfamethoxazole (SMX) is a known potentiator of DHFR inhibitors that acts by inhibiting the synthesis of the DHFR substrate, dihydrofolic acid[39]. The combination of fluorofolin and SMX exhibited significant synergy in

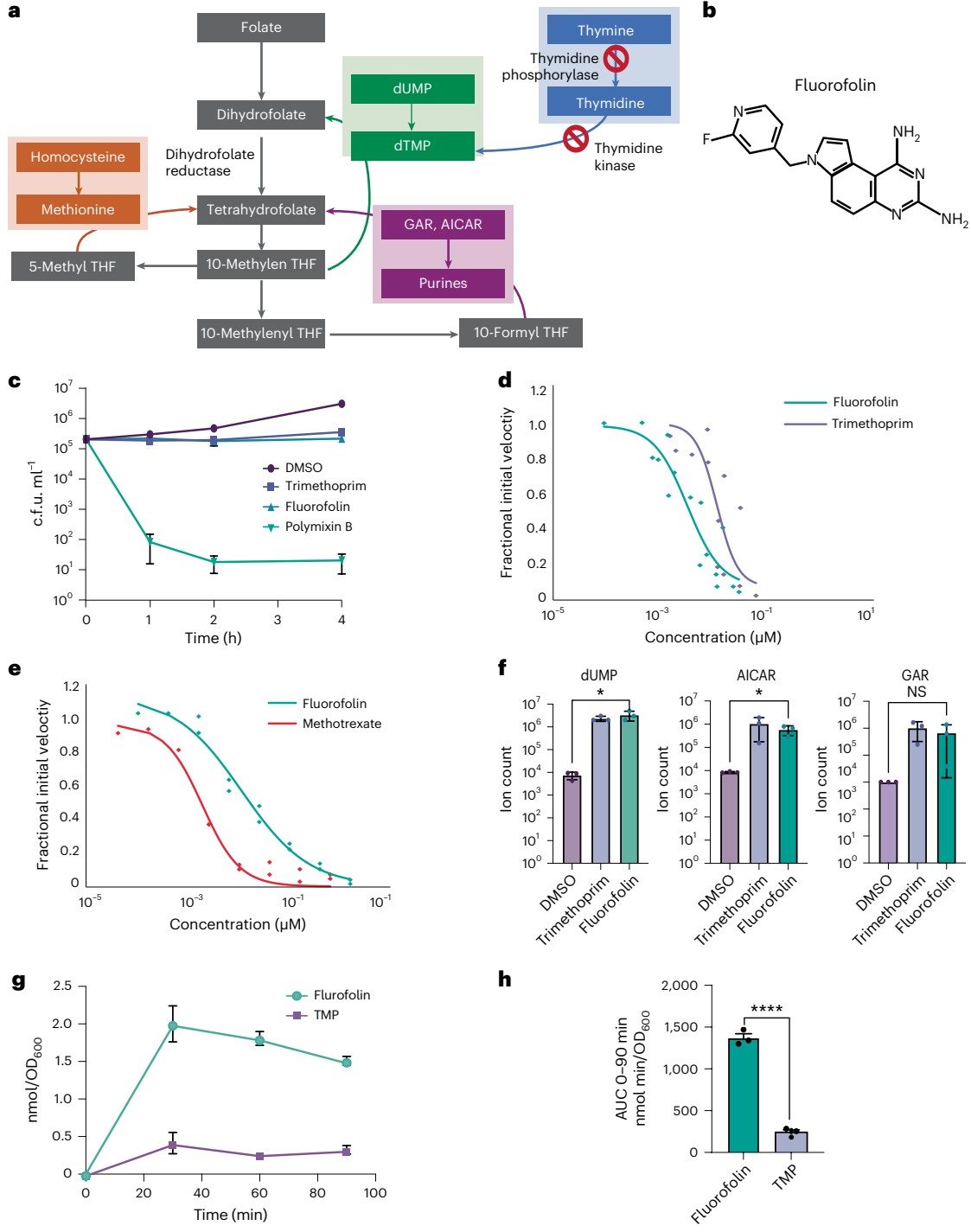

**Fig. 1 | Fluorofolin inhibits *P. aeruginosa* growth through inhibition of DHFR and rapidly accumulates within cells. a**, Schematic of folate metabolism in PA14. Red crossed circles indicate enzymes lacking activity in PA14. **b**, The structure of fluorofolin. **c**, C.f.u.s ml⁻¹ of *P. aeruginosa* PA14 after 4 h treatment with 5% dimethylsulfoxide (DMSO, solvent control), 6.2 µg ml⁻¹ fluorofolin (2× MIC), 250 µg ml⁻¹ TMP (2× MIC) or 4 µg ml⁻¹ polymixin B (2× MIC). Data points represent 3 biological replicates with 3 technical replicates. Mean ± s.d. **d**, DHFR (FolA) activity measured on purified *E. coli* FolA by measuring the change in sample absorbance at 340 nm due to DHFR-dependent NADPH consumption. Activity was related to an untreated standard condition using 60 µM NADPH and 100 µM DHF. IC₅₀ values were derived from the Hill equation fits on reactions performed with increasing antibiotic concentrations. **e**, An analogous assay to

Fig. 2b using purified human DHFR. **f**, Metabolite abundance of dUMP, AICAR and glycinamide ribonucleotide (GAR) of *P. Aeruginosa* PA14 treated with 5% DMSO (solvent control), 6.3 µg ml⁻¹ fluorofolin (2× MIC) or 250 µg ml⁻¹ TMP (2× MIC) for 15 min. Metabolite abundance was quantified in comparison to the solvent-only control. Data represent mean ± s.d. for 3 biological replicates. *$P \leq 0.05$; NS, not significant; calculated using two-sided unpaired *t*-test using Prism 9. **g**, Intrabacterial drug metabolism (IBDM) of PA14 treated with 5 µM fluorofolin or TMP. Samples were taken at time $t = 0$, 30, 60 and 90 min. Data represent mean ± s.d. of 3 technical replicates. **h**, Cumulative accumulation of drugs over 90 min by AUC of IBDM curves. Data represent mean ± s.d. of 3 technical replicates. ****$P < 0.0005$, $P = 4.5512 \times 10^{-5}$; two-sided *t*-test using Prism 9.

PA14 (Fig. 2b). Specifically, in the presence of well-tolerated doses of SMX (156 µg ml⁻¹), the MIC of fluorofolin was reduced to 0.4 µg ml⁻¹ (Fig. 2b), establishing a promising therapeutic window with respect to concentration of fluorofolin achievable in vivo.

To investigate the ability of fluorofolin to clear *P. aeruginosa* infection in vivo, we utilized a murine thigh infection model of PA14. We administered fluorofolin subcutaneously at the maximum tolerated dose (25 mg kg⁻¹) along with a dose of SMX that has been previously established to be clinically relevant[40] (100 mg kg⁻¹ intraperitoneally) and measured PA14 load after 24 h. The combination of fluorofolin and SMX significantly inhibited the growth ($P < 0.0001$) of PA14 compared with both no treatment and SMX alone (Fig. 2c). To support the feasibility of thymidine supplementation in vivo, we performed an additional mouse *P. aeruginosa* infection model in which we included a group of mice fed with a thymidine-supplemented diet starting at 2 days before infection. This group also showed a significant reduction in *P. aeruginosa* after 24 h compared with untreated mice (Fig. 2d).

## Fluorofolin selectively targets *P. aeruginosa* with exogenous thymine

Supplementation of media with thymine, methionine and inosine (TMI) can rescue DHFR-mediated growth inhibition by TMP in *E. coli*[41]. We confirmed that fluorofolin inhibition of *E. coli* MG1655 growth could also be rescued with TMI supplementation (Fig. 3a); however, TMI supplementation was unable to rescue *P. aeruginosa* PA14 from fluorofolin treatment (Fig. 3b). These results are consistent with fluorofolin functioning as a broad-spectrum DHFR inhibitor and *P. aeruginosa* lacking the enzymes required for utilizing exogenous thymine. To test the hypothesis that inhibiting DHFR in the presence of thymine would selectively inhibit the growth of *P. aeruginosa*, we co-cultured *E. coli* MG1655 and *P. aeruginosa* PA14 in the presence of TMI supplementation. In these conditions, fluorofolin selectively inhibited the growth of *P. aeruginosa* but not *E. coli*, as quantified by plating on selective media (Fig. 3c). In addition to the c.f.u. counts, the selective killing of *P. aeruginosa* was evident in thymine-supplemented liquid cultures as *P. aeruginosa* cultures exhibit a characteristic blue-green hue from pyocyanin production that disappeared upon fluorofolin treatment (Fig. 3d). To demonstrate that the narrow-spectrum nature of fluorofolin is not specific to *E. coli*, we demonstrated that other species including *S. epidermidis*, *E. cloacae*, *E. faecalis*, *K. pneumoniae* and *S. aureus* could also be rescued from fluorofolin growth inhibition by TMI supplementation (Extended Data Fig. 3a–e). Rescue from fluorofolin inhibition was also shown in *E. coli lptd34213* using thymidine supplementation alone (Extended Data Fig. 3f), as well as in the presence of relevant concentrations of SMX (Extended Data Fig. 3g). These results support the conclusion that *P. aeruginosa* PA14 is unusual in its inability to salvage exogenous thymine and demonstrate that thymine supplementation can convert fluorofolin from a broad-spectrum antibiotic to a narrow-spectrum antibiotic.

## Fluorofolin resistance attenuates *P. aeruginosa* virulence

One of the hallmarks of the fluorofolin parent compound, SCH-79797, was a lack of observable resistance due to a dual-targeting mechanism of action[26]. Because fluorofolin acts solely as a DHFR inhibitor, we hypothesized that resistance to fluorofolin could more readily occur. We were indeed able to isolate two different fluorofolin-resistant mutants of *P. aeruginosa* PA14. One type of fluorofolin-resistant mutant was isolated by plating 10⁸ cells onto LB agar plates containing 4× MIC fluorofolin. Resistance frequency on these plates was 1 in $1.5 \times 10^6$ cells. While this mutation frequency is high, whole-genome sequencing of these resistance mutants revealed that all the mutants mapped to a singular protein-coding gene, *nfxB*. Of the 8 mutants sequenced, 6 had a T39P point mutation, 1 had an L29R point mutation and 1 had a premature stop codon at amino acid 115.

**Table 1 | MIC of fluorofolin against a panel of bacteria**

| Species | Strain | MIC (µg ml⁻¹) |
|---|---|---|
| *Enterococcus faecalis* | ATCC 51575* | 0.01 |
| *Escherichia coli* | lptD4213 | 0.02 |
| *Salmonella typhimurium* | CMCC 50115* | 0.03 |
| *Staphylococcus aureus* | NRS384* | 0.03 |
| *Escherichia coli* | NCM-3722 | 0.3125 |
| *Staphylococcus epidermidis* | EGM-206 | 0.3125 |
| *Staphylococcus aureus* | USA300 | 0.4 |
| *Escherichia coli* | MG1655 | 0.625 |
| *Acinetobacter baumannii* | ATCC 17978* | 0.8 |
| *Pseudomonas aeruginosa* | PA14 | 3.125 |
| *Pseudomonas aeruginosa* | PA01 | 6.25 |
| *Klebsiella pneumoniae* | ATCC 438165 | 6.25 |
| *Escherichia coli* | ATCC BAA-198* | 7.11 |
| *Enterobacter cloacae* | ATCC BAA-1143 | 12.5 |
| *Pseudomonas aeruginosa* | ATTC 27853 | 25 |
| *Acinetobacter baumannii* | BAA-125 | 31.25 |

MIC represents the concentration of fluorofolin at which no bacterial growth is detected after 16 h at 37 °C in LB broth as measured by OD₆₀₀. MIC of strains denoted with * were performed by WuXi in Mueller–Hinton broth. MIC values were calculated from duplicate samples.

We confirmed fluorofolin resistance of *nfxB* mutants in liquid culture (MIC > 100 µg ml⁻¹).

NfxB is a transcriptional regulator protein that represses expression of the MexCD-OprN efflux pump[7]. We confirmed that our *nfxB* mutants cause upregulation of MexCD-OprN using RNA-seq (Fig. 4a). *P. aeruginosa nfxB* mutants have also been shown to confer resistance to other antibiotics, including ciprofloxacin[37]. As expected, we confirmed that the fluorofolin-resistant *nfxB* T39P mutants were also cross-resistant to ciprofloxacin (Fig. 4b). Since the same mutations can confer resistance to fluorofolin and ciprofloxacin, we determined whether the two antibiotics also have similarly high resistance frequency in our resistance plating assay. Resistance to ciprofloxacin was even more frequent than that of fluorofolin ($P = 0.042$) (Fig. 4c). Fluorofolin also had significantly lower resistance frequency than meropenem ($P = 0.0342$) and similar levels of resistance to gentamycin. These resistance frequencies are higher than expected of a clinically used antibiotic such as ciprofloxacin, but below we demonstrate that they probably do not represent the resistance frequency observed in vivo.

Our animal infection study suggested that fluorofolin should be co-administered with SMX (Fig. 2). We thus examined the rate of resistance to fluorofolin–SMX combination treatment. To do so, we plated 10⁹ c.f.u.s on 10 plates containing 4× MIC of fluorofolin and the lowest concentration of SMX at which we observed synergy (78.1 µg ml⁻¹). In these conditions, we were unable to isolate resistance mutants, indicating that the resistance frequency is below our level of detection (<1 in 10¹⁰).

The other class of fluorofolin-resistant mutant isolated arose through serial passaging of *P. aeruginosa* PA14 at 0.5× MIC fluorofolin for 10 passages. Whole-genome sequencing revealed that the only protein-coding mutation in these mutants was a point mutation in *mexS* L46F (MIC = 62.5 µg ml⁻¹). MexS is an oxidoreductase that represses MexT, which in turn induces expression of another efflux pump, MexEF-OprJ[42]. We confirmed that our *mexS* mutant caused upregulation MexEF-OprJ (Fig. 4d) using RNA-seq.

We next explored the physiological impacts of the *nfxB* and *mexS* efflux pump upregulation mutants. The *nfxB* and *mexS* mutants had significantly decreased production of the quorum sensing phenazine pyocyanin ($P = 0.0001$ and $P = 0.0006$, respectively) compared with

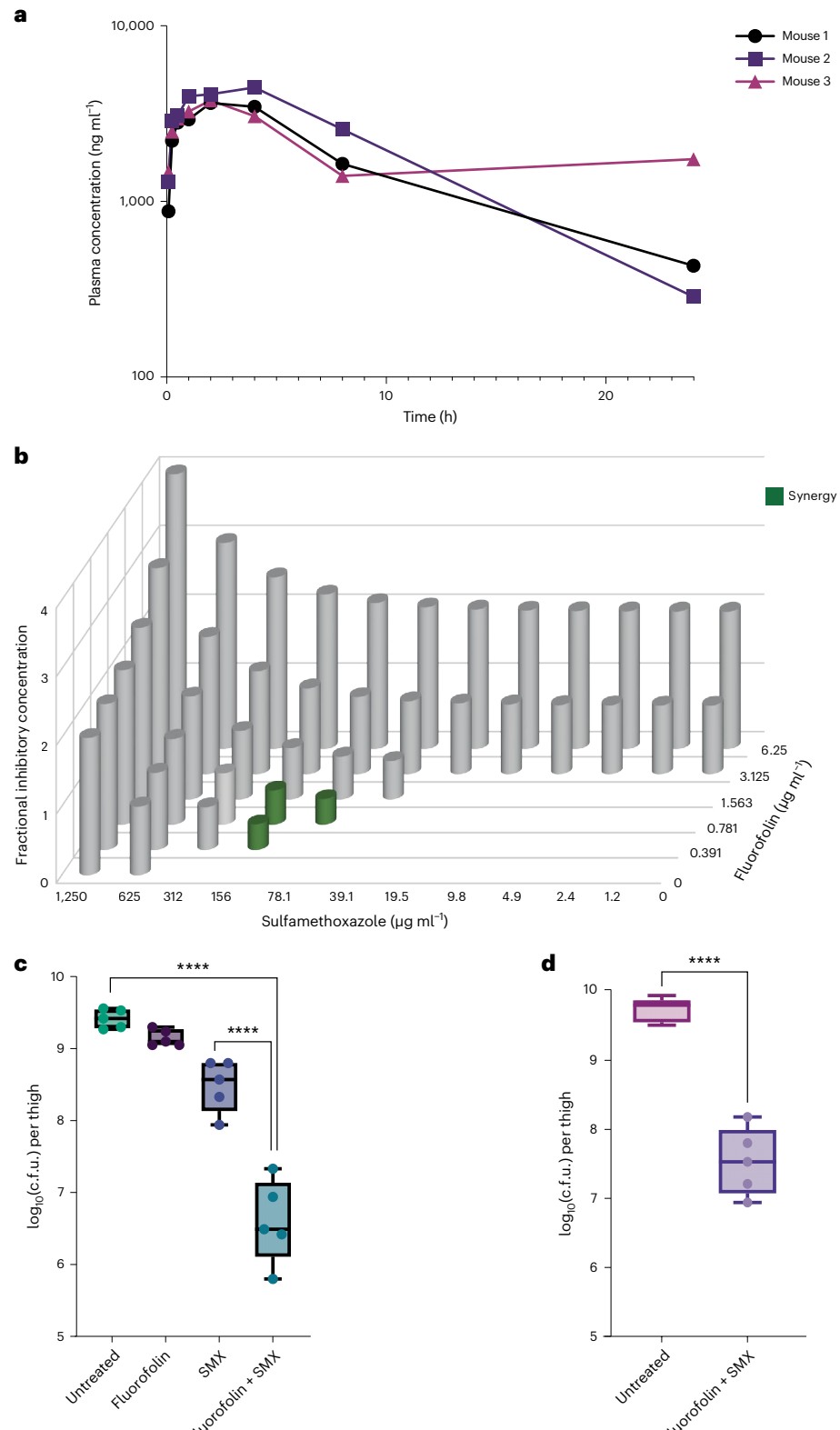

**Fig. 2 | In vivo efficacy of fluorofolin against *P. aeruginosa* PA14. a,** Plasma concentration of fluorofolin over time after single oral administration to neutropenic CD-1 mice. Each line is representative of an individual mouse. **b,** Checkerboard assay of fluorofolin and sulfamethoxazole. *Z*-values represent FICs determined by dividing the MIC of each drug when used in combination by the MIC when used alone. A FIC ≤ 0.5 is considered a synergistic effect. **c,** Treatment of mice with fluorofolin (subcutaneous) with or without SMX (100 mg kg⁻¹, intraperitoneal). Mice were treated at 1 and 12 h post infection (*n* = 5 for each group). ****$P \leq 0.0001$ from two-sided Tukey's multiple

comparisons test. Boxplot whiskers range from minima to maxima. Boxplots extend from the 25th to the 75th percentiles, with a line drawn at the median value for each group. **d,** Fluorofolin and SMX treatment of mice fed a diet of thymidine-supplemented chow during PA14 infection. Mice were treated at 1 and 12 h post infection (*n* = 5 for each group). ****$P \leq 0.0001$ from two-sided Tukey's multiple comparisons test. Boxplot whiskers range from minima to maxima. Boxplots extend from the 25th to the 75th percentiles, with a line drawn at the median value for each group.

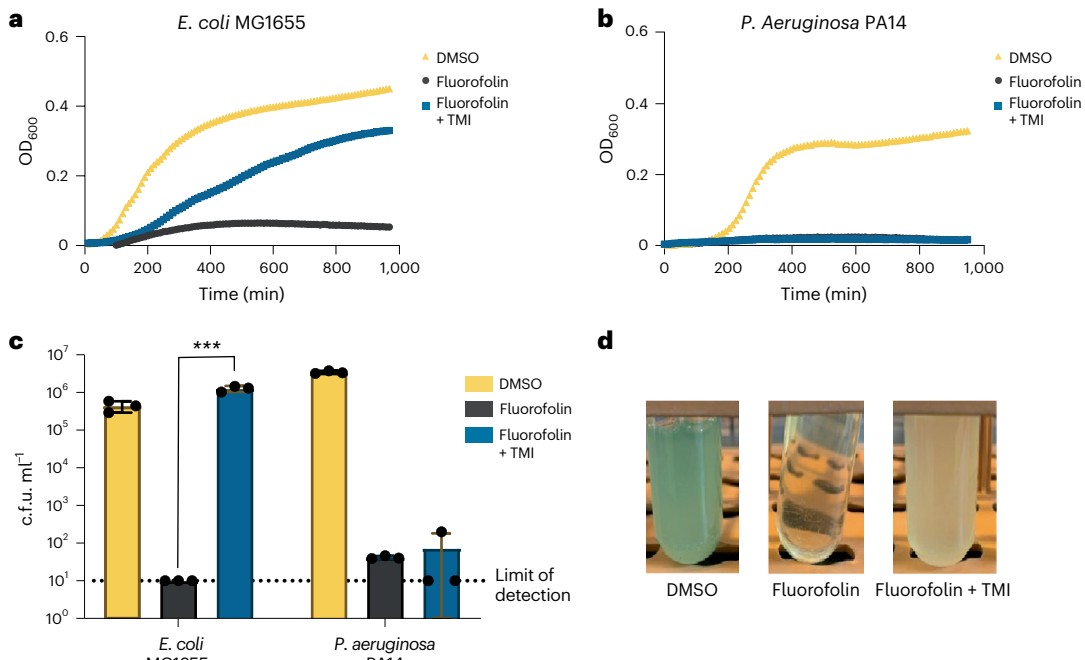

**Fig. 3 | Divergent thymidine metabolism can be exploited using thymine supplementation to specifically target *P. aeruginosa* with fluorofolin.**
**a**, Growth of *E. coli* MG1655 or **b**, *P. aeruginosa* PA14 in 0.3 mM TMI-supplemented media and treated with fluorofolin at 2× MIC or DMSO. Curves represent average OD$_{600}$ of 2 biological replicates. **c**, Competition of *P. aeruginosa* PA14 and *E. coli* MG1655 in LB or TMI-supplemented LB media. PA14 and *E. coli* MG1655 were inoculated at a 1:1 ratio and grown overnight in the presence or absence of 50 µg ml⁻¹ fluorofolin in each media condition. The following day, c.f.u.s were counted on LB agar or *Pseudomonas* selection agar plates to determine c.f.u. ml⁻¹ of each species. Data represent mean ± s.d. of 3 biological and 3 technical replicates. ***$P \leq 0.001$, $P = 0.0006$; calculated using two-sided unpaired $t$-test using Prism 9. **d**, Representative culture images.

wildtype PA14 (Fig. 4e). Efflux pump overexpression could increase secretion of quorum sensing precursors, thereby inhibiting the accumulation of the quorum sensing molecules themselves. We found that pyocyanin production could be partially rescued by addition of the quorum sensing molecule, *Pseudomonas* quinolone signal (PQS) (Fig. 4e), which is known to both induce pyocyanin and have precursors that are susceptible to efflux[43,44]. Quorum sensing and pyocyanin are known to affect *P. aeruginosa* virulence, suggesting that the fluorofolin-resistant mutants could also affect *P. aeruginosa* pathogenesis. Both fluorofolin-resistant mutants were significantly less virulent than WT PA14 ($P < 0.0001$) in a *C. elegans* N2 infection model (Fig. 4f), which correlates well with mammalian pathogenicity[45]. These results suggest that there is a trade-off between fluorofolin resistance and virulence in *P. aeruginosa*.

## Fluorofolin resistance is limited in clinical isolates

If there is a trade-off between fluorofolin resistance and virulence in vivo, we would predict that most clinical isolates of *P. aeruginosa*, which are known to be virulent, should be susceptible to fluorofolin[6]. To investigate the predominance of fluorofolin resistance in clinical *P. aeruginosa* isolates, we obtained the CDC and FDA Antibiotic Resistant *P. aeruginosa* Isolate Bank[46] (strains 1–55) and 22 clinical isolates from blood stream infections (strains 56–77)[47]. Only 10.4% of these clinical isolates were resistant to fluorofolin at 50 µg ml⁻¹. None of the 22 blood isolates and only 8 of the 55 CDC isolates (which are specifically enriched for antibiotic-resistant isolates) were resistant to fluorofolin. The resistance frequency to fluorofolin compared favourably with the clinically used antibiotics ciprofloxacin, gentamycin and meropenem, which had resistance frequencies of 15.6%, 35.0% and 74.0%, respectively, in the same strains (Fig. 4g). While no folate inhibitors are in clinical use against *P. aeruginosa*, we found that all of the eight fluorofolin-resistant clinical isolates in our collection had a multidrug

resistance cassette containing a *dfrB5* integron with a mutant allele of DHFR known to confer clinical resistance to TMP in *Klebsiella pneumoniae*[46]. This result confirms DHFR as the physiological target of fluorofolin in vivo and demonstrates that while DHFR mutations can result in fluorofolin resistance, such mutations are rare among existing *P. aeruginosa* clinical isolates. Finally, we found that not all TMP-resistant DHFR mutations confer cross-resistance with fluorofolin, as strains 7, 18 and 23 of the CDC panel are fluorofolin-sensitive despite having *dfrB2*, another TMP-resistant DHFR allele.

Importantly, while our *nfxB* efflux pump overexpression strain showed cross-resistance to both fluorofolin and ciprofloxacin (Fig. 4b), none of the *P. aeruginosa* isolates in our collection demonstrated cross-resistance to both antibiotics (Fig. 4f). This result supports our hypothesis that in *P. aeruginosa*, efflux pump upregulation readily confers antibiotic resistance in vitro, but that such mutations rarely accumulate in clinical settings. We were also encouraged to observe that *P. aeruginosa* clinical isolates expressing *KPC-5* and *NDM-1* carbapenemase genes, which are considered by the CDC to be critical targets of antibiotic research and development[48], remained sensitive to fluorofolin. Thus, while the rate at which fluorofolin resistance might emerge upon treatment in vivo remains to be determined, fluorofolin appears to be effective against most existing *P. aeruginosa* clinical isolates, including those of significant clinical interest.

## Discussion

*P. aeruginosa* is a leading cause of nosocomial infections for which antibiotic development is urgently needed. *P. aeruginosa* infections are typically first treated with the fluroquinolone, ciprofloxacin[49]. However, more than 40% of *P. aeruginosa* clinical isolates are reported to be resistant to ciprofloxacin, diminishing clinical options[49]. Furthermore, ciprofloxacin and other antibiotics currently used for *P. aeruginosa* (such as piperacillin-tazobactam) are broad-spectrum, disrupting the host microbiome in a manner that often does not fully recover

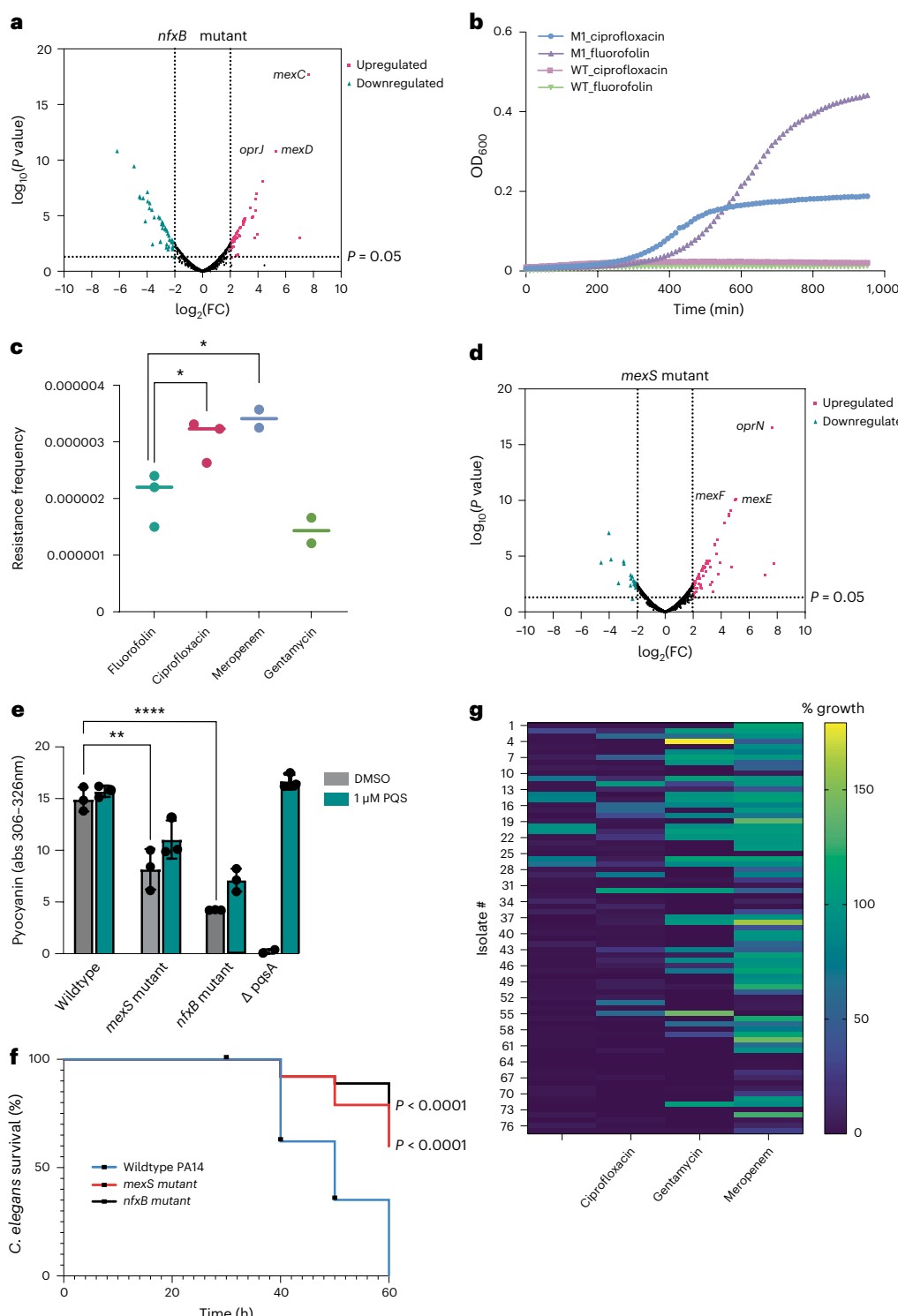

**Fig. 4 | Fluorofolin mutants overexpress efflux pumps and have decreased virulence. a**, RNA sequencing results from an *nfxB* (T39P) mutant relative to wildtype PA14. *P* values calculated using two-tailed quasi-likelihood *F*-test. **b**, *nfxB* (T39P) mutants show cross-resistance to both ciprofloxacin (2× MIC) and fluorofolin (2× MIC). Curves represent average OD₆₀₀ of 2 biological replicates. **c**, Resistance frequency of 4× MIC of a panel of antibiotics tested against PA14. Points represent independent plates. *\**P* < 0.05; calculated using two-sided unpaired *t*-test using Prism 9. **d**, RNA sequencing results from a *mexS* (L46F) mutant relative to wildtype PA14. *P* values calculated using two-tailed quasi-likelihood *F*-test. **e**, Pyocyanin production of *nfxB* (T39P) and *mexS* (L46F) mutants. Pyocyanin levels were measured by integration of absorbances from 306 to 326 nm. A Δ*pqsA* PA14 mutant was included as this strain does not make pyocyanin and Δ*pqsA* absorbance values were used to subtract out background

signal. PQS in DMSO (1 μM) was added to samples at inoculation to rescue pyocyanin production. Data are presented as mean ± s.d. of biological triplicates for each group. *\*\*\**P* < 0.0001, *\**P* < 0.01; calculated using two-sided unpaired *t*-test using Prism 9. **f**, *C. elegans* N2 toxicity after infection with wildtype PA14 (*n* = 158), *nfxB* T39P (*n* = 131) or *mexS* L46F (*n* = 128) was measured over 60 h. Worms were declared dead if they lacked movement after being gently poked with forceps. *P* values were calculated using a Mantel–Cox test compared to wildtype PA14 using Prism 9. **g**, Clinical isolate resistance to fluorofolin and ciprofloxacin was tested by treating the panel with 50 μg ml⁻¹ of either antibiotic. Growth inhibition was determined by comparing OD₆₀₀ after 16 h to DMSO-treated controls. Strains with growth inhibition >80% were considered sensitive to fluorofolin treatment.

after treatment[12,50]. There is consequently an urgent need for compounds that are both narrow spectrum for *P. aeruginosa* and effective against existing multidrug-resistant strains.

Here we identify an antibiotic compound, fluorofolin, capable of inhibiting the growth of *P. aeruginosa* through potent DHFR inhibition. Fluorofolin is effective both in vitro and in a mouse thigh infection model. Fluorofolin represents the first folate inhibitor that is effective at tolerated doses in *P. aeruginosa*. Fluorofolin's activity in *P. aeruginosa* highlights the well-known but often overlooked point that target engagement is not the only important factor to consider in developing antibiotics. The IC$_{50}$ of fluorofolin and TMP against *E. coli* DHFR are comparable in vitro. However, fluorofolin has superior accumulation in *P. aeruginosa*, allowing it to better access its target within the bacteria. Designing antibiotics on the basis of their capacity to be effluxed by the MexAB-OprM efflux pump may also be a viable strategy to revive antibiotics with no intrinsic activity against *P. aeruginosa*.

Fluorofolin's ability to target *P. aeruginosa* also enabled us to demonstrate that *P. aeruginosa*, which lacks the ability to scavenge exogenous thymine present in most other bacterial species, can be selectively targeted by DHFR inhibition in the presence of thymine supplementation. While resistance to fluorofolin is possible, common mechanisms of resistance also confer a decrease in bacterial virulence and most clinical isolates of *P. aeruginosa* are susceptible to fluorofolin. The clinical isolates resistant to fluorofolin all express a mutant DHFR, *dfrB5*. The spread of *dfrB5* among clinical isolates could be a potential concern for the clinical development of fluorofolin. However, as TMP has not been used to treat *P. aeruginosa* infection, we predict that these alleles are not very widespread among existing *P. aeruginosa* isolates, and it is possible that co-treatment with SMX will mitigate this issue.

Recent studies have highlighted the importance of microbiome integrity for multiple aspects of human health, including metabolic and immune system regulation[51,52]. Broad-spectrum antibiotics can disrupt the microbiome leading to dysbiosis, highlighting the need for more targeted antibiotic approaches[15,53]. Traditional efforts to develop narrow-spectrum antibiotics have focused on targeting features that are specifically present in a bacterial species of interest. Here, however, we exploit the specific absence of a thymine salvage pathway in *P. aeruginosa* to show that a broad-spectrum inhibitor of folate synthesis can selectively inhibit the growth of this important pathogen through thymine supplementation. Most human commensal bacteria have thymine kinase homologues[22,23] and we confirmed that multiple bacterial species' sensitivity to fluorofolin can be rescued by thymine supplementation. We note that a few other bacterial species also lack thymine kinase, including the human pathogens *Helicobacter pylori*[22,54] and *M. tuberculosis*[23], suggesting that these pathogens could also be selectively targeted using a similar approach. Actinomycetes and their closely related genera *Corynebacterium, Mycobacterium* and *Rhodococcus* have also been shown to lack thymidine kinase activity[23], but these bacteria represent a small subset of those present in the human microbiome and are predominantly found within skin communities[55]. In humans, thymidine supplementation has been shown to be safe and is routinely used to reduce toxicity associated with methotrexate treatment[56].

In contrast to its parent molecules, SCH-79797 and IRS-16 (ref. [26]), fluorofolin lacks the ability to disrupt bacterial membranes. This divergence in activity improves the therapeutic index of fluorofolin, probably due to reduced off-target effects on mammalian membranes. However, this change in mechanism of action also allows for resistance against fluorofolin to develop more easily. In vitro, we were able to isolate two fluorofolin-resistant mutants, which were both attributed to the overexpression of efflux pumps (MexCD-OprJ in one mutant and MexEF-OprN in the other). MexCD-OprJ overexpression has been shown to confer resistance to cefpirome and quinolones[6], while MexEF-OprN overexpression has been shown to confer resistance to imipenem, chloramphenicol and quinolones[42].

While fluorofolin-resistant mutants that overexpress efflux pumps could be readily isolated in vitro, we also find that these mutants have significantly reduced virulence, which would explain their low frequency in pathogenic clinical isolates. These findings are consistent with studies from *P. aeruginosa* PAO1 demonstrating that overexpression of efflux pumps decreases type III secretion[57], secretion of virulence factors[58] and swarming[8]. MexCD-OprJ has been suggested to efflux 2-heptyl-4-quinolone (HHQ)[59] while MexEF-OprN has been suggested to efflux kynurenine[44]. HHQ and kynurenine are precursors of PQS, a key molecule in regulating *Pseudomonas* quorum sensing and virulence[60]. Together, these data suggest that high efflux pump levels secrete quorum sensing precursors, preventing the synthesis of quorum sensing molecules that promote virulence and thereby reducing virulence. In support of this hypothesis, we demonstrated that addition of exogenous PQS was able to partially restore pyocyanin production to the fluorofolin-resistant mutants. Finally, mutants in MexCD-OprJ are hypersusceptible to imipenem[61] while mutants in MexEF-OprN are hypersusceptible to aminoglycoside and β-lactams[42], such that combination therapies of fluorofolin with traditional antibiotics may also prove effective at addressing any residual resistance to fluorofolin. It may also be possible to further modify the fluorofolin scaffold to select for derivatives with decreased propensity to be effluxed by these pumps.

The prevalence of efflux pump upregulation in *P. aeruginosa* clinical isolates is an important subject that will require additional investigation. Among the clinical isolates examined in this study, we did not observe any cross-resistance to both ciprofloxacin and fluorofolin, suggesting that there was no efflux pump upregulation. These findings are consistent with a different study reporting that mutants that upregulate efflux pumps are rarely isolated from *P. aeruginosa* clinical samples[6]. However, there are also reports of elevated efflux pump expression in *P. aeruguniosa* isolates from cystic fibrosis patients[62]. It remains unclear whether this difference is specific to cystic fibrosis or whether there are other contexts in which efflux pump mutants survive in the host. To address scenarios in which efflux pump upregulation does prove to be prevalent, it will be interesting to explore the potential benefits of antibiotic cycling or combinations as a means to exploit these mutants' collateral sensitivity to other antibiotics[63,64].

## Methods

### Experimental models and subject details

**Bacterial strains and growth conditions.** Bacterial strain information is provided in Table 1. Where listed, growth media were prepared according manufacturer recommendations: LB broth and LB broth supplemented with 0.3 mM thymine (BD Biosciences 244610, Alfa Aesar A15879), 0.3 mM methionine (Sigma-Aldrich M9625), 0.3 mM inosine (EMD Millipore 4060), cation-adjusted Mueller–Hinton II broth (CAMHB) (BD 212322) or 0.3 mM thymidine (Sigma-Aldrich T1895), Gutnick minimal media (1.0 g l$^{-1}$ K$_2$SO$_4$, 13.5 g l$^{-1}$ K$_2$HPO$_4$, 4.7 g l$^{-1}$ KH$_2$PO$_4$, 0.1 g l$^{-1}$ MgSO$_4$·7H$_2$O, 10 mM NH$_4$Cl as a nitrogen source and 0.4% w/v glucose as a carbon source)[65].

**Animal models.** For pharmacokinetics determination, care and handling of male CD-1 mice approximately 6–8 weeks old conformed to institutional animal care and use policies as carried out at Pharmaron.

For *P. aeruginosa* thigh infection model and maximum tolerated dose (MTD) studies, care and handling of female 5–6-week-old CD-1 conformed to Institutional Animal Care and Use Committee (IACUC) policies as carried out at the University of North Texas Health Science Center (Fort Worth, Texas) under UNTHSC IACUC-approved protocol nos. IACUC-2021-0003 and IACUC-2020-0039. Rodents were fed either base chow (Envigo) or base diet supplemented with 1.8 g kg$^{-1}$ thymidine (Sigma T9250) starting at 2 days before infection. All procedures were conducted in accordance with the UNTHSC IACUC-approved protocol. Animals were housed in rooms undergoing 10–15 air changes per hour.

Air provided to the animal rooms was controlled for temperature and humidity and is fully monitored 24 h a day. Air pressure was balanced according to room use. Lighting was controlled in individual rooms by automatic timers with a standard 12 h on, 12 h off cycle.

## MIC results

MIC was defined as the lowest concentration of antibiotic at which no visible growth was detected after 16 h at 37 °C. Overnight cultures were diluted 1:150 in LB broth and added to a 96-well plate. Antibiotics were serially diluted 1:2 and added to columns of the 96-well plate and grown at 37 °C with continuous shaking. Cell growth was measured using optical density at 600 nm ($OD_{600}$). MIC assays were performed in either BioTek Synergy HT or Tecan InfiniteM200 Pro microplate readers.

For MIC calculations performed by WuXi, MIC was calculated as the lowest concentration that inhibits visible growth after 18 h. Bacterial colonies (4–8) of strains of interest were vortexed in saline and adjusted to an $OD_{600}$ of 0.2. Strains were diluted 1:200 into CAMHB media in 96-well plates. Antibiotics were serially diluted 1:3 in DMSO, and 1 µl of each dilution was added to bacteria. The plates were incubated for 18–20 h at 37 °C before observation.

## Colony-forming units counting

*P. aeruginosa* PA14 overnight cultures were diluted 1:100 and grown to exponential phase ($OD_{600} = 0.4–0.6$). Cultures were diluted 1:10 and treated with antibiotic. At each time point, 150 µl of culture was removed, serial diluted 1:10 six times and plated onto LB agar plates. Plates were grown overnight at 37 °C, after which visible colonies were counted. C.f.u.s ml$^{-1}$ are reported from dilutions in which ~10–100 single colonies were visible.

## Membrane potential and permeability assay

Overnight *P. aeruginosa* PA14 or *E. coli lptD4213* cultures were diluted 1:100 and grown to mid exponential phase at 37 °C. Cultures were diluted 1:10 into PBS and treated with antibiotics for 15 min. *P. aeruginosa* PA14 was stained with TO-PRO-3 (640 nm excitation, 670/30 nm emission) to measure cell membrane integrity. *E. coli lptD4213* was stained with both TO-PRO-3 and DiOC2(3) (ThermoFisher B34950) to measure cell membrane integrity and membrane potential. DiOC2(3) was evaluated as a ratio of green (488 nm excitation, 525/50 nm emission) to red (488 nm excitation, 610/20 nm emission)[66]. The LSRII flow cytometer (BD Biosciences) at the Flow Cytometry Resource Facility, Princeton University, was used to measure the fluorescent intensities of both dyes in response to antibiotic treatment. 100,000 events were recorded for each data file. Gates for permeabilization were determined using polymixin B (Sigma-Aldrich P1004) and untreated controls. Gates for depolarization were determined using carbonyl cyanide m-chlorophenyl hydrazone (CCCP) as a positive control. Data were analysed using FlowJo v.10 software.

## MexAB-OprM transposon mutants

*P. aeruginosa* PA14 transposon mutants were generated by the Ausubel Lab (https://pa14.mgh.harvard.edu/cgi-bin/pa14/home.cgi)[67]. The MICs of fluorofolin and TMP against strains with disrupted MexA, MexB and OprM were determined as above and compared to the parental strain. As transposon mutants in MexB were represented twice in this collection, the MIC was confirmed across both mutant strains.

## Haemolysis

Defibrinated sheep red blood cells (Lampire 50414518) were diluted to $6 \times 10^6$ cells per ml, pelleted and washed 3× with PBS. Samples were treated at 37 °C with shaking for 1 h and then centrifuged. Supernatants were collected and absorbances were measured at 405 nm in a Tecan InfiniteM200 Pro microplate reader. Percentage haemolysis was calculated compared to 100% lysis by Triton X-100 (1% v/v) (Sigma-Aldrich X100RS).

## Mammalian cell cytotoxicity

HK-2 (500 cells per well) (ATCC CRL-2190), HLF (500 cells per well) (Cell Applications 506K-05a), WI-38 (500 cells per well) (ATCC CCL-75) or PBMC (5,000 cells per well) (TPCS PB010C) cells were seeded in white opaque 384-well plates. After 24 h, DMSO or compounds were added in 3-fold dilutions and cells incubated for 72 h. For PBMC, CellTiter-Glo reagent was added in equal volume and incubated for 30 min, after which luminescence was read. For other cell types, CyQUANT detection reagent was added in equal volume and cells incubated for 1 h, after which fluorescence was read with standard green filter set (508/527 nm excitation). Cell toxicity was evaluated by Pharmaron.

## Metabolomics

Overnight *P. aeruginosa* PA14 cultures were diluted 1:150 in Gutnick minimal media and grown to early–mid exponential phase ($OD_{600} = 0.4–0.6$). Cultures were treated with either 6.3 µg ml$^{-1}$ fluorofolin (2× MIC) or 250 µg ml$^{-1}$ trimethoprim (2× MIC) (Chem-Impex 01634) for 15 min. Metabolites were extracted by vacuum filtering 15 ml of treated cells using 0.45 µm HNWP Millipore nylon membranes and placing the filters into an ice-cold quenching solution of 40:40:20 methanol:acetonitrile:$H_2O$. Extracts were kept on dry ice for 1 h and centrifuged at 16,000 g for 1 h at 4 °C. The supernatant was kept at −80 °C until mass spectrometry analysis.

Liquid chromatography–mass spectrometry (LC–MS) analysis of metabolites was performed on an Orbitrap Exploris 240 mass spectrometer coupled with hydrophilic interaction liquid chromatography (HILIC)[68]. HILIC was on an XBridge BEH Amide column (2.1 mm × 150 mm, 2.5 µM particle size; Waters 196006724), with a gradient of solvent A (95 vol. % $H_2O$, 5 vol. % acetonitrile, with 20 mM ammonium acetate and 20 mM ammonium hydroxide, pH 9.4) and solvent B (acetonitrile). Flow rate was 0.15 ml min$^{-1}$ and column temperature was set at 25 °C. The LC gradient was: 0–2 min, 90% B; 3–7 min, 75% B; 8–9 min, 70% B; 10–12 min, 50% B; 12–14 min, 25% B; 16–20.5 min, 0.5% B; 21–25 min, 90% B. The orbitrap resolution was 180,000 at an $m/z$ of 200. The maximum injection time was 200 ms and the automatic gain control target was 1,000%. Raw mass spectrometry data were converted to mzXML format by MSConvert (ProteoWizard). Pick-peaking was done on El Maven (v.0.8.0, Elucidata).

## In vitro DHFR *E. coli*

As described previously[26], purified *E. coli* dihydrofolate reductase (FolA) was purified by Genscript. Enzyme activity was measured on a QuantaMaster 40 spectrophotometer (Photon Technology) using the DHFR reductase assay kit with slight modifications. *E. coli* FolA was diluted 1:1,000 into 1× assay buffer. Of this mixture, 100 µl with or without compound was added to a BRAND UV cuvette (Sigma-Aldrich BR759200) and sample transmitted light intensity at 340 nm was measured for 100 s at 1 kHz sampling. Readings were averaged for every 1 Hz and the activity of each sample was calculated from the slope ($\beta$) of a linear regression of the log-transformed intensity measurements on MATLAB R2022B. To account for enzyme stability, measurements were normalized to a standard condition (60 µM NADPH and 100 µM DHF) measured immediately before the sample of interest. The relative activity was calculated as $(\beta_{\text{sample}} - \beta_{\text{noEnzyme}})/(\beta_{\text{standard}} - \beta_{\text{noEnzyme}})$.

## Human DHFR in vitro assay

Human purified DHFR was purchased from R&D Systems (8456-DR). DHFR activity was assayed by monitoring the decrease in absorbance by NADPH at 340 nM. DHFR enzyme (0.5 µg ml$^{-1}$), dihydrofolic acid (100 µM) and different concentrations of methotrexate, fluorofolin or DMSO control were dissolved in 200 µl of Tris buffer (pH 7.5, Tris salt concentration 25 mM). Reaction was initiated by adding NADPH in a 10× stock (1 mM for a final concentration of 100 µM), and absorbance at 340 nM was monitored over time using a Cytation 5 reader (Agilent). Activity was normalized to the DMSO control.

## Molecular docking

As the structure of *P. aeruginosa* DHFR has not been solved, we used AlphaFold[30,31] to derive the enzyme's three-dimensional structure from its sequence (UniProt ID: 6XG5)[32]. Following the acquisition of the protein structure, we introduced the coenzyme NADPH to the structure, aligning it on the experimentally characterized human DHFR in a complex with IRS-17, providing a structural reference for subsequent steps.

The structures of fluorofolin and trimethoprim were translated from SMILES representation using the RDKit cheminformatic package. After preparing the enzyme, coenzyme and ligand structures, we defined a cubic grid box of dimensions $20 \times 20 \times 20$ Å centred around the active site of the reference human DHFR–IRS-17 complex. This box serves as the search space for potential binding sites in our docking simulations. We executed the docking simulation using the Auto-Dock Vina[33,34] forcefield, with an exhaustiveness parameter set to 64 to ensure comprehensive sampling of the search space.

## Checkerboard assay

Cells were seeded in a similar manner as described above for MIC calculations. Sulfamethoxazole (Chem-Impex 00821) was diluted 1:2 down the rows of the plate, while fluorofolin was diluted 1:2 down the columns of the plate. Fractional inhibitory concentrations (FICs) were calculated as [fluorofolin]/$MIC_{Fluorofolin}$ + [SMX]/$MIC_{SMX}$, where [fluorofolin] and [SMX] are the concentrations of compounds in a given well, which were divided by the concentration of drug at the MIC for each compound. FIC values less than or equal to 0.5 were considered synergistic.

## Growth competition assay

Overnight cultures of *P. aeruginosa* PA14 or *E. coli* MG1655 were diluted 1:150 into LB broth in the presence of DMSO or 50 µg ml$^{-1}$ fluorofolin with or without TMI supplementation and grown for 16 h at 37 °C. Cultures of each species were grown separately as well as being mixed 1:1. Cultures were plated onto LB agar or *Pseudomonas* Selection Agar (Sigma-Aldrich 17208) and c.f.u. counting was caried out as described above. To control for appropriate *Pseudomonas* selection, *E. coli* MG1655 was plated onto *Pseudomonas* Selection Agar and an absence of colonies was observed. The number of colonies on *Pseudomonas* Selection Agar plates is reported as the c.f.u. ml$^{-1}$ of *P. aeruginosa*. To calculate c.f.u.s ml$^{-1}$ of *E. coli* MG1655, c.f.u.s ml$^{-1}$ were determined from LB agar plates and the c.f.u.s ml$^{-1}$ of *P. aeruginosa* were subtracted from these values.

## Drug accumulation assay

Overnight PA14 cultures were back-diluted to early–mid exponential phase ($OD_{600} = 0.4$–$0.6$). The assay was initiated with treatment of the culture with either 5.0 µM fluorofolin or 5.0 µM trimethoprim. A DMSO-treated culture was utilized as a control. At time points of 30, 60 and 90 min, a 10 ml aliquot was collected out of the 120 ml parent culture (in triplicate) and pelleted by centrifugation at 2,000 $g$ at 4 °C. The supernatant was then removed and the pellet was washed with ice-cold 0.85% NaCl solution. Following suspension of the cell pellet in 1 ml of 2:2:1 $CH_3CN$:MeOH:$H_2O$, samples were subjected to four cycles of freeze–thaw cell lysis using dry ice in 95% ethanol/ice water. Before each freeze phase, samples were vortexed for 10 s to ensure adequate mixing. Samples were subsequently pelleted at 16,000 $g$ for 5 min, with the supernatant being subjected to filtration using a 0.22 µm SpinX centrifuge tube filter. The resulting cell lysate samples were analysed, utilizing verapamil as an internal standard. For LC–MS analysis, sample components were separated using a Chromolith SpeedRod column, using a gradient of 10–100% $CH_3CN$/$H_2O$ acidified with 0.1% (v/v) formic acid, with an Agilent 1260 liquid chromatograph coupled to an Agilent 6120 quadruple mass spectrometer.

Compound accumulation was realized using the selective ion monitoring mode to quantify peak integration for a compound and the internal standard using their respective *m/z* values. Compound peaks were confirmed using a scanning mode that detected the compound peak using an *m/z* range of 100–1,000. Peak area integration values were determined and a ratio of the peak area for the compound to the peak area for the verapamil internal standard was calculated, and compound concentration was then determined from the compound calibration curve. The calculated concentration of the compound in each sample was then normalized using the bacterial culture $OD_{600}$ value. Compound accumulation versus time plots were generated using GraphPad Prism v.9.4.1. Compound accumulation area under the curve (AUC, calculated in Microsoft Excel v.16.65) was determined for each bacterial strain–compound combination and these were compared via statistical analysis (unpaired *t*-test) in GraphPad Prism v.9.4.1.

## Fluorofolin resistance screens

For resistance passaging, *P. aeruginosa* PA14 was grown overnight at 37 °C in a 96-well plate similarly to MIC assays in duplicate. The wells corresponding to 0.5× MIC was selected and struck out on LB agar plates in the absence of antibiotic to select for stable resistance. Single colonies were picked and inoculated into fresh LB broth. This process was repeated for a total of ten passages. At each passage, the MIC was recalculated and compared to a culture that had not been previously exposed to fluorofolin, which was also grown as a control to confirm antibiotic potency. Cells from each passage were stored as a frozen stock.

To isolate resistant mutants on a plate, $10^8$ c.f.u.s of *P. aeruginosa* PA14 were plated on LB agar plates containing 4× MIC fluorofolin, ciprofloxacin, meropenem or gentamycin. We plated $10^9$ c.f.u.s on 10 plates containing 4× MIC fluorofolin with 4× the minimal concentration of SMX at which synergy was observed. Plates were grown at 37 °C for 48 h, after which individual colonies were picked and restruck onto fresh plates and grown at 4× MIC of the relevant antibiotic LB broth to confirm resistance. Resistant mutants were maintained as a frozen stock.

To confirm the identity of resistance mutations, whole-genome sequencing was performed and compared to the parental strain of PA14. Briefly, genomic DNA was isolated from a strain of interest using the DNeasy blood and tissue kit (Qiagen 69504). Once DNA was extracted and its quality was confirmed, the DNA was sequenced using an Illumina NextSeq 2000 system. Sequencing and variant calling were performed at SeqCenter.

## RNA sequencing

RNA was extracted from overnight cultures of wildtype PA14 or mutant strains. Cultures were pelleted at 4 °C, resuspended in Trizol (Ambion 10296010) and incubated at 65 °C before addition of chloroform. The aqueous layer was collected and RNA was isolated using a mirVana RNA extraction kit (ThermoFisher AM1560). DNAse treatment was performed on RNA using recombinant DNase I (Sigma-Aldrich 04716728001). RNA samples were sent to SeqCenter for sequencing using the Illumina Stranded RNA library preparation with RiboZero Plus rRNA depletion kit.

## Pharmacokinetic analysis

Pharmacokinetic properties were determined after a single dose of 200 mg kg$^{-1}$ fluorofolin given orally (PO). Plasma samples were taken from three mice at times 0.083, 0.25, 0.5, 1, 2, 4, 8 and 24 h, and quantitative analysis was performed using LC–MS/MS. Half-life was determined from plasma concentration after fluorofolin levels reached pseudo-equilibrium (4 h for mouse 1 and 2, and 2 h for mouse 3). Pharmacokinetic values were estimated using a non-compartmental model generated from WinNonlin 6.1. Pharmacokinetic analysis was carried out by Pharmaron.

## Serum binding

Fluorofolin (1 µM) was added to a mouse plasma solution or to a buffer-only control. An initial $t = 0$ sample was collected. Fluorofolin

was incubated with plasma for 6 h, after which dialysis was performed. After dialysis, supernatant was collected and the amount of fluorofolin was determined by LC–MS/MS to determine the percentage of unbound fluorofolin. Serum binding parameters were determined as follows:

1. % Bound = 100% − % Unbound
2. $\log K = \log(\%\ \text{Bound}/100 − \%\ \text{Bound})$
3. % Remaining = Area-ratio$_{6h}$/Area-ratio$_{0h}$ × 100%
4. % Recovery was determined as (Area-ratio$_{\text{buffer-chamber}}$ + Area-ratio$_{\text{plasma-chamber}}$)/(Area-ratio$_{\text{total sample}}$) × 100

Serum binding analysis was performed by Pharmaron.

### *C. elegans* maintenance and toxicity assay

*C. elegans* N2 worms were maintained on *E coli* OP50-coated nematode growth medium (NGM) plates before experiments. For *P. aeruginosa*-coated plates, overnight cultures were diluted to OD$_{600}$ = 1, spread onto NGM plates, incubated overnight at 37 °C and equilibrated to 25 °C. To synchronize worms for virulence assays, young adult hermaphrodites were bleached to obtain eggs and synchronized L4 worms were collected at 2 days post bleaching. For virulence assays, synchronized L4 worms were transferred to *P. aeruginosa* plates. Worms were counted at time $t$ = 0, 30, 40, 50 and 60 h to assess viability. Worms were declared dead if they lacked movement after being gently poked with forceps. *P* values were calculated using a Mantel–Cox test in Prism 9 to compare mutant virulence to wildtype PA14 virulence.

### Clinical isolate panels

Clinical isolates[46] were inoculated into LB broth in a 96-well flat-bottom plate and grown overnight to stationary phase at 37 °C. The following day, strains were diluted 1:150 into fresh LB broth with fluorofolin or ciprofloxacin at 50 µg ml$^{-1}$ or vehicle control wells and incubated at 37 °C overnight. Percent growth was calculated by dividing the OD$_{600}$ of fluorofolin antibiotic-treated wells by that of vehicle-only wells.

### Pyocyanin production

Overnight cultures of PA14, PA14 Δ*pqsA*[69] and each mutant were grown in biological triplicate and the OD$_{600}$ of the cultures were measured. Cell-free supernatants were calculated by centrifugation, and 100 µl were added to a 96-well plate in duplicate. The integrated absorbance spectrum from 306 to 326 nm was taken in a Tecan InfiniteM200 Pro plate reader to determine pyocyanin levels in each sample. A Δ*pqsA* sample was used to subtract out any background values, and pyocyanin levels were normalized by the ratio of OD$_{600}$ between wildtype PA14 and the mutants to account for any slight differences in growth.

### Maximum tolerated dose

MTD was determined by administration of compounds at increasing dosage until the maximum dose before adverse reactions were observed. Doses were increased in a stepwise manner from 1 mg kg$^{-1}$ to 5, 10, 25 and 50 mg kg$^{-1}$. Mice were observed for adverse effects including respiration, piloerection, startle response, skin colour, injection site reactions, hunched posture, ataxia, salivation, lacrimation, diarrhoea, convulsion and death. The maximum tolerated dose of fluorofolin was determined to be 25 mg kg$^{-1}$ administered subcutaneously. MTD was evaluated by the UNTHSC.

### In vivo PA14 infection

Female 5–6-week-old CD-1 mice were rendered neutropenic by intraperitoneal cyclophosphamide treatment (Cytoxan) before this study. On day 0, mice were infected intramuscularly with 5.33 log c.f.u. per thigh PA14. Mice were treated subcutaneously with fluorofolin at 1 and 12 h post infection. For co-treatment, mice were treated with SMX intraperitoneally at 1 and 12 h post infection. Mice were euthanized using CO$_2$ at 24 h post infection, after which thighs were removed and placed into sterile PBS, homogenized and serially diluted onto brain

heart infusion (BHI) and charcoal plates for c.f.u. counting. In vivo efficacy was evaluated by the UNTHSC.

### Statistical information

For all assays showing error bars, we used the arithmetic mean and standard deviation across multiple biological replicates as our measures of centre and spread. The number of replicates for each experiment type and the type of statistical test used to determine significance are included in respective figure legends.

### Reporting summary

Further information on research design is available in the Nature Portfolio Reporting Summary linked to this article.

## Data availability

The datasets generated during and/or analysed during the current study are available from the corresponding author on reasonable request. RNA sequencing data have been deposited at GEO (accession GSE249862). Metabolomics data have been deposited at https://massive.ucsd.edu/ProteoSAFe/QueryMSV?id=MSV000093598. Source data are provided with this paper.

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

## Acknowledgements

Fluorofolin was supplied by Y. Huang and C. Jiang (Princeton-PKU Shenzhen). Flow cytometry was performed with the assistance of C. DeCoste (Princeton University Flow Cytometry Resource Facility (FCRF)). *C. elegans* were a generous gift from C. Murphy's lab (Princeton University). *S. epidermidis* strain was obtained from L. Flowers and E. Grice (University of Pennsylvania). We appreciate the support and feedback from members of the Gitai and Shaevitz labs. Funding was provided in part by Rutgers/NJACTS Pilot Program (23307-E4289-10012776-FA781 to Z.G.) and for the FCRF by the National Cancer Institute (NCI-CCSG P30CA072720-5921) as well as the Princeton Intellectual Property Accelerator Program (to Z.G.). The opinions, findings, and conclusions or recommendations expressed in this material contents are solely the responsibility of the authors and do not necessarily represent the official views of the funding sources.

## Author contributions

Z.G., C.C., J.P.S. and H.K. conceptualized the project. Z.G., C.C., J.P.S. and J.S.F. developed the methodology. C.C. and J.P.S. performed validation. C.C. and J.P.S. conducted formal analysis. C.C., J.P.S., A.G., X.X. and S.G. conducted investigations. Z.G. and C.C. wrote the original draft. Z.G., C.C., J.P.S., H.K. and J.S.F. reviewed and edited the manuscript. C.C. and J.P.S. performed visualization. Z.G., H.K., J.D.R. and J.S.F. supervised the project. Z.G. and J.D.R. acquired funding.

## Competing interests

A patent application (PCT/US2023/076235 Princeton, NJ, 2023) for the use of Fluorofolin as an antibiotic is currently pending. Z.G. is a co-founder and has equity interest from Arrepath. The remaining authors declare no competing interests.

## Additional information

**Extended data** is available for this paper at https://doi.org/10.1038/s41564-024-01665-2.

**Correspondence and requests for materials** should be addressed to Zemer Gitai.

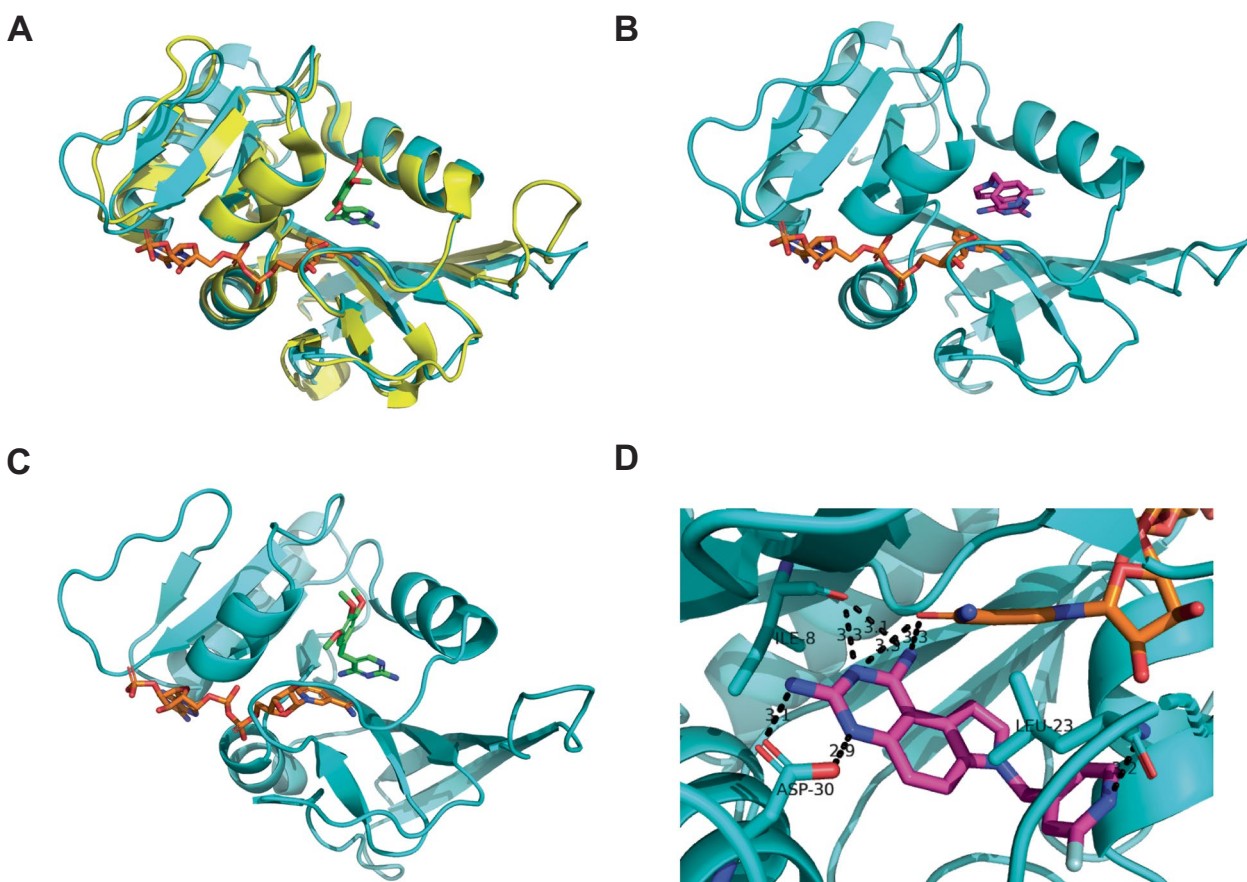

**Extended Data Fig. 1 | Fluorofolin predicted binding to P. aeruginosa DHFR.** (**a**) Superimposition PA DHFR on 6XG5 to locate the binding site. PA DHFR in cyan, 6XG5 in yellow, NADPH in orange, and TMP in green (**b**) Molecular docking of fluorofolin with PA DHFR and the cofactor NADPH. The figure shows the lowest-scoring pose as predicted by AutoDock Vina. Fluorofolin, PA DHFR and NADPH are shown in pink, cyan and orange, respectively. (**c**) Molecular docking of TMP with PA DHFR and the cofactor NADPH. The figure shows the lowest-scoring pose as predicted by AutoDock Vina. TMP, PA DHFR and NADPH are shown in green, cyan and orange, respectively. (**d**) Polar interaction between fluorofolin with the neighboring residues LEU-23, ILE-8 and ASP-30. All figures are rendered using PyMOL[70].

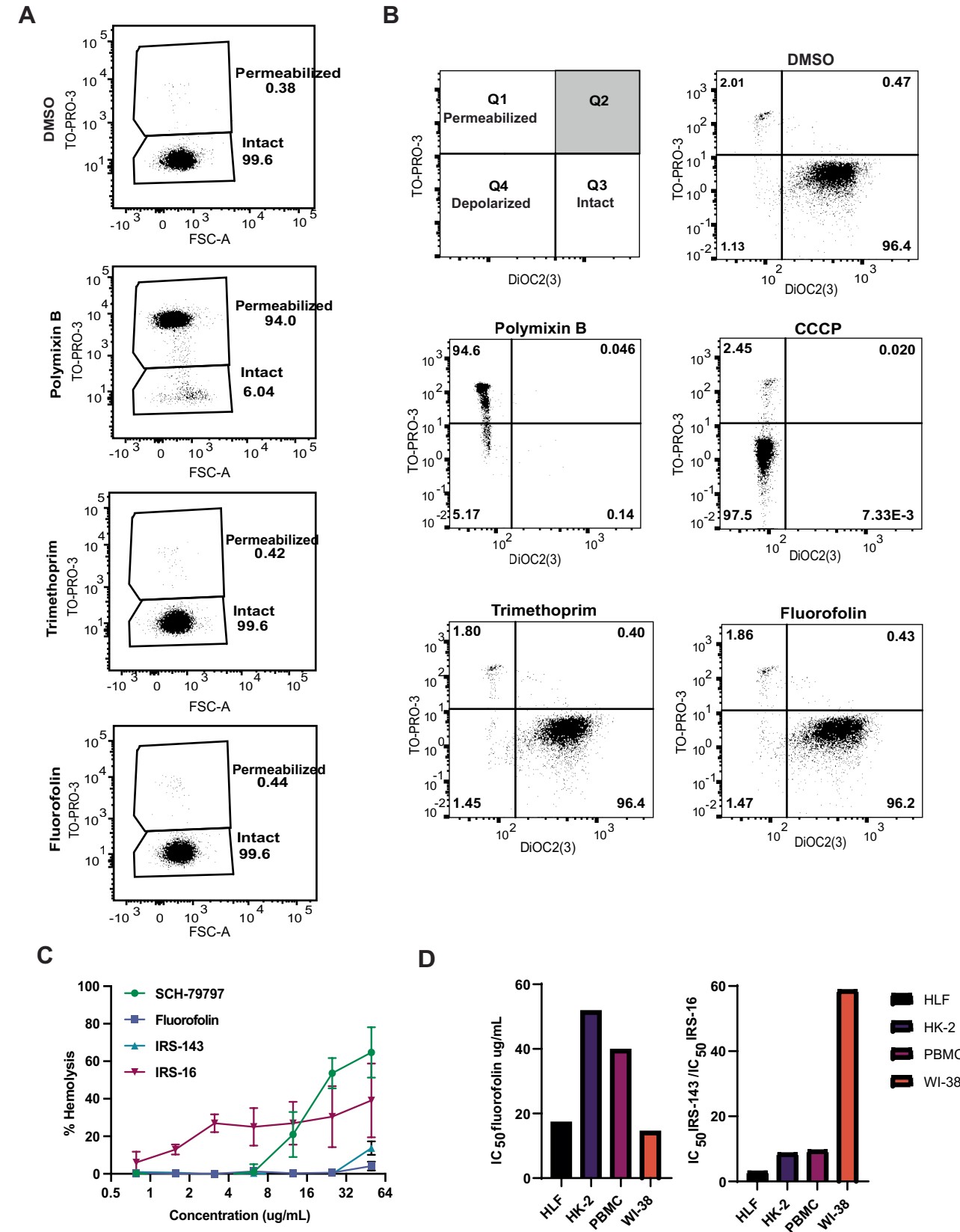

**Extended Data Fig. 2 | See next page for caption.**

**Extended Data Fig. 2 | Fluorofolin does not cause membrane permeabilization or depolarization improving therapeutic index.** (**a**) Flow cytometry results of PA14 stained with the membrane permeability dye TO-PRO-3. Cells were incubated for 15 min with 5% DMSO (solvent control), 4 µg/mL polymixin B (2X MIC), 250 µg/mL TMP (2X MIC), or 6.25 µg/mL fluorofolin (2X MIC). The gates were determined for TO-PRO-3 staining using solvent only and polymixin B controls. (**b**) Flow cytometry *of E. coli lptD4213* treated with 5% DMSO, 5 µM CCCP, or 2x MIC fluorofolin 0.04 µg/mL polymixin B, and 0.4 µg/mL TMP and stained with TO-PRO-3 and DiOC2(3). (**c**) Hemolysis of $6 \times 10^6$ sheep red blood cells after treatment with selected antibiotics for 1 hour. Percent hemolysis was measured using $Abs_{405}$ compared to 100% lysis control by Triton X-100 (1% v/v). Mean ± SD of technical triplicates are shown. (**d**) IC50 of fluorofolin against *in vitro* mammalian cell lines relative to the IC50 of IRS-16. HLF: human lung fibroblast, HK-2: human kidney epithelial, PBMC: peripheral blood mononuclear cell, WI-38: Embryonic lung tissue.

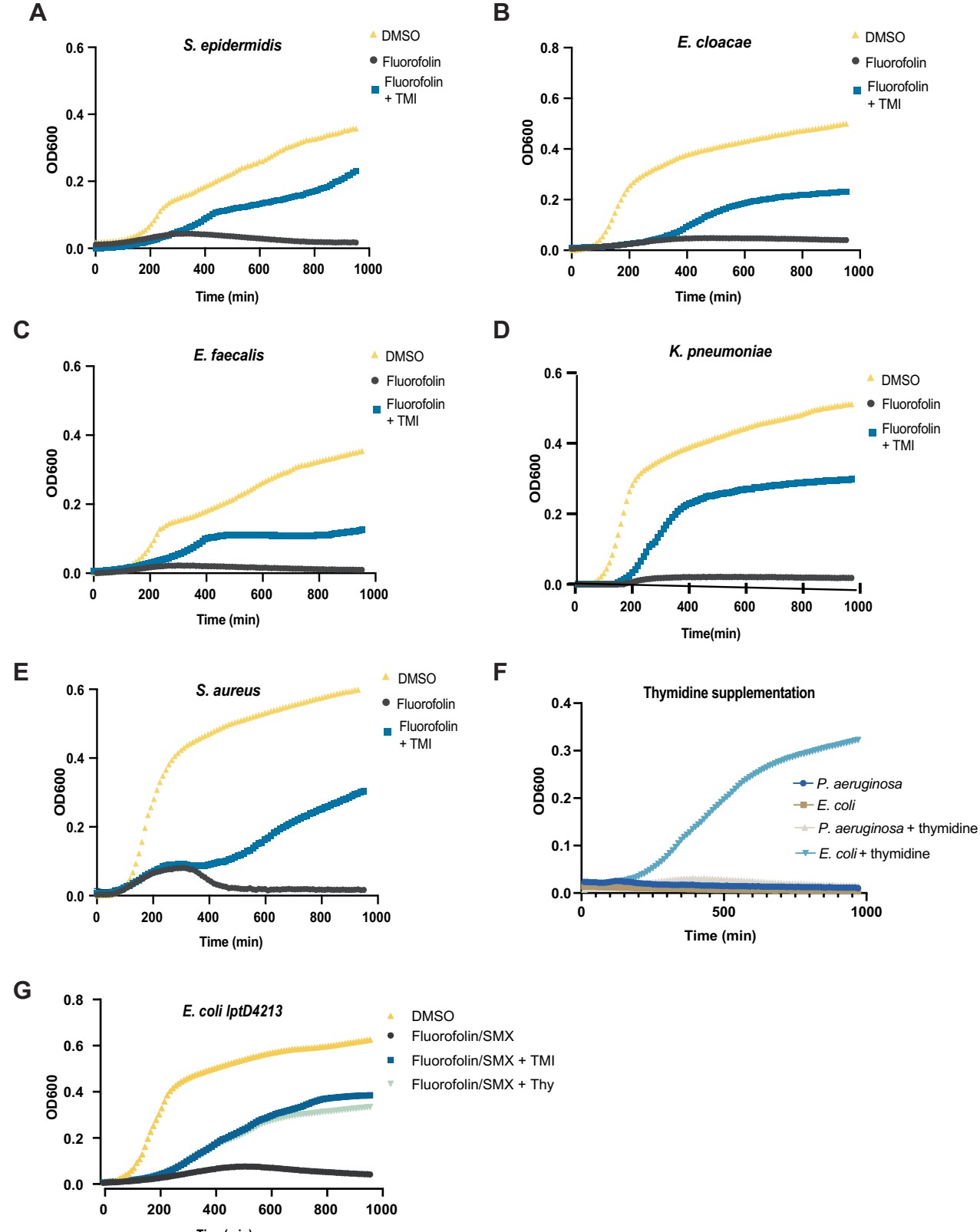

**Extended Data Fig. 3 | TMI supplementation can also rescue other species from fluorofolin DHFR inhibition.** Growth of (**a**) S. epidermidis (**b**) E. cloacae (**c**) E. faecalis (**d**) K. Pneumoniae (**e**) S. aureus USA300 with or without TMI supplementation (0.3 mM thymine, methionine, and inosine) and treated with fluorofolin at 2X MIC or DMSO. (**f**) P. aeruginosa PA14 or E. coli lptd4213 treated with fluorofolin 50 μg/mL in the presence or absence of 0.3 mM thymidine. (**g**) E. coli lptd4213 treated with fluorofolin 50 μg/mL and SMX (78.1 μg/mL) in the presence or absence of 0.3 mM thymidine or 0.3 mM TMI supplementation, Curves represent average optical at 600 nm (OD$_{600}$) of 2 biological replicates. Means are shown.

**Extended Data Table 1 | Minimum inhibitory concentration of fluorofolin or TMP against transposon mutants for each component of the MexAB-OprM efflux pump**

| MIC (µg/mL) | Trimethoprim | Fluorofolin |
|---|---|---|
| mexA | 50 | 1.5 |
| mexB | 25 | 1.5 |
| oprM | 25 | 1.5 |
| Wildtype | 125 | 3.1 |

MIC against wildtype PA14 was calculated to control for each drug stock. MIC values are representative of two independent replicates.

**Extended Data Table 2 | Pharmacokinetic (PK) and plasma binding (PB) parameters of fluorofolin**

| Pharmacokinetic Parameter | Mean | SD | Unit |
|---|---|---|---|
| T1/2 | 12.1050 | 10.5370 | h |
| Tmax | 2.6670 | 1.1550 | h |
| Cmax | 3953.3330 | 442.8690 | ng/mL |
| AUC_last | 46333.9500 | 6520.7810 | h*ng/mL |
| AUC_Inf | 68752.7840 | 34235.1190 | h*ng/mL |
| AUC_% Extrap_obs | 23.3970 | 28.7420 | % |
| MRT_Inf_obs | 18.2170 | 17.0420 | h |
| AUC_last/D | 231.6700 | 32.6040 | h*ng/mL |
| | | | |
| **Plasma Binding Parameter** | | | |
| % Bound | 71.7000 | 2.3550 | % |
| % Unbound | 28.3000 | 2.3550 | % |
| LogK | 0.4000 | 0.0490 | |
| % Recovery | 91.8800 | 5.6070 | % |
| % Remaining after 6 hours | 89.7200 | 1.8310 | % |

PK values were calculated from three mice given 200 mg/kg fluorofolin PO. Plasma was collected at 0.083, 0.25, 0.5, 1, 2, 4, 8, and 24 hours post treatment for LC/MS-MS analysis. PB values were determined from incubation of fluorofolin with purified mouse plasma over 6 hours. PB parameters were calculated from LC/MS-MS measurements of duplicate samples.

# Reporting Summary

## Statistics

For all statistical analyses, confirm that the following items are present in the figure legend, table legend, main text, or Methods section.

| n/a | Confirmed | |
|---|---|---|
| ☐ | ☒ | The exact sample size (*n*) for each experimental group/condition, given as a discrete number and unit of measurement |
| ☐ | ☒ | A statement on whether measurements were taken from distinct samples or whether the same sample was measured repeatedly |
| ☐ | ☒ | The statistical test(s) used AND whether they are one- or two-sided<br>*Only common tests should be described solely by name; describe more complex techniques in the Methods section.* |
| ☐ | ☒ | A description of all covariates tested |
| ☐ | ☒ | A description of any assumptions or corrections, such as tests of normality and adjustment for multiple comparisons |
| ☐ | ☒ | A full description of the statistical parameters including central tendency (e.g. means) or other basic estimates (e.g. regression coefficient) AND variation (e.g. standard deviation) or associated estimates of uncertainty (e.g. confidence intervals) |
| ☐ | ☒ | For null hypothesis testing, the test statistic (e.g. *F*, *t*, *r*) with confidence intervals, effect sizes, degrees of freedom and *P* value noted<br>*Give P values as exact values whenever suitable.* |
| ☒ | ☐ | For Bayesian analysis, information on the choice of priors and Markov chain Monte Carlo settings |
| ☒ | ☐ | For hierarchical and complex designs, identification of the appropriate level for tests and full reporting of outcomes |
| ☒ | ☐ | Estimates of effect sizes (e.g. Cohen's *d*, Pearson's *r*), indicating how they were calculated |

*Our web collection on statistics for biologists contains articles on many of the points above.*

## Software and code

Policy information about availability of computer code

| Data collection | No software was used. |
|---|---|
| Data analysis | GraphPad Prism Version 9.4.1., FlowJo v10, and MATLAB R2022b were used for data analysis. |

For manuscripts utilizing custom algorithms or software that are central to the research but not yet described in published literature, software must be made available to editors and reviewers. We strongly encourage code deposition in a community repository (e.g. GitHub). See the Nature Portfolio guidelines for submitting code & software for further information.

## Data

Policy information about availability of data

All manuscripts must include a data availability statement. This statement should provide the following information, where applicable:
- Accession codes, unique identifiers, or web links for publicly available datasets
- A description of any restrictions on data availability
- For clinical datasets or third party data, please ensure that the statement adheres to our policy

The datasets generated during and/or analyzed during the current study are available from the corresponding author on reasonable request. Links to raw data for metabolomics and RNA sequencing depositories are provided in the final submission of the manuscript (GEO (GSE249862) and MassIVE (MSV000093598)).

## Human research participants

Policy information about studies involving human research participants and Sex and Gender in Research.

| | |
|---|---|
| Reporting on sex and gender | N/A |
| Population characteristics | N/A |
| Recruitment | N/A |
| Ethics oversight | N/A |

Note that full information on the approval of the study protocol must also be provided in the manuscript.

# Field-specific reporting

Please select the one below that is the best fit for your research. If you are not sure, read the appropriate sections before making your selection.

☒ Life sciences        ☐ Behavioural & social sciences        ☐ Ecological, evolutionary & environmental sciences

For a reference copy of the document with all sections, see nature.com/documents/nr-reporting-summary-flat.pdf

# Life sciences study design

All studies must disclose on these points even when the disclosure is negative.

| | |
|---|---|
| Sample size | Sample size calculations were not performed. Number of replicates in reported in relevant figures. For C. elegans experiments, n were chosen based on similiar stuides in the literature (Kaletsky et al 2020). Sample size for mouse experiments was chosen based on previous mouse infection model experiments performed at University of North Texas Health Science Center. For murine pharmacokinetic experiments, n=3 was used for determination of drug properties. For murine infection experiments, n=5 per group was used. |
| Data exclusions | C. elegans data was excluded from this study if a worm could not be located on a given plate (this worm would be censored from any subsequent analysis). Only worms that were confirmed dead or alive were included in the analysis. All other data was included in this study. |
| Replication | Experiments were repeated in biological duplicate or triplicate and, at minimum, in technical duplicate to confirm reproducibility. All attempts at replication were successful. Animal experiments were performed using multiple mice in randomly allocated groups. |
| Randomization | Organisms were allocated to each group randomly. |
| Blinding | Blinding was not relevant to this study as all outcomes were objective measurements and therefore were not biased by labeled groups. |

# Reporting for specific materials, systems and methods

We require information from authors about some types of materials, experimental systems and methods used in many studies. Here, indicate whether each material, system or method listed is relevant to your study. If you are not sure if a list item applies to your research, read the appropriate section before selecting a response.

### Materials & experimental systems

| n/a | Involved in the study |
|---|---|
| ☒ | ☐ Antibodies |
| ☐ | ☒ Eukaryotic cell lines |
| ☒ | ☐ Palaeontology and archaeology |
| ☐ | ☒ Animals and other organisms |
| ☒ | ☐ Clinical data |
| ☒ | ☐ Dual use research of concern |

### Methods

| n/a | Involved in the study |
|---|---|
| ☒ | ☐ ChIP-seq |
| ☐ | ☒ Flow cytometry |
| ☒ | ☐ MRI-based neuroimaging |

## Eukaryotic cell lines

Policy information about cell lines and Sex and Gender in Research

| | |
|---|---|
| Cell line source(s) | HLF: human lung fibroblast (Cell Applications 506K-05a), HK-2: human kidney epithelial (ATCC-CRL2190), PBMC: peripheral |

| Cell line source(s) | blood mononuclear cell (TPCSP-B010C), WI-38: Embryonic lung tissue (ATC-CCL-75) provided by Pharmaron, Inc. (Beijing, ROC). |
|---|---|
| Authentication | Authentication of cell lines was performed by Pharmaron, Inc. (Beijing, ROC) using STR profiling. |
| Mycoplasma contamination | Mycoplasma testing was performed by Pharmaron, Inc. (Beijing, ROC). All cell lines tested were negative for mycoplasma. |
| Commonly misidentified lines (See ICLAC register) | None |

# Animals and other research organisms

Policy information about studies involving animals; ARRIVE guidelines recommended for reporting animal research, and Sex and Gender in Research

| Laboratory animals | Male CD-1 mice aged 6–8 weeks old and female CD-1 mice aged 5–6-weeks old were used in this study. All procedures were conducted in accordance with a protocol approved by the UNTHSC Animal Care and Use Committee (IACUC). Animals were housed with rooms undergoing ten to fifteen air changes per hour. Air provided to the animal rooms was controlled for temperature and humidity and is fully monitored, 24 hours a day. Air pressures was balanced according to room use. Lighting was controlled in individual rooms by automatic timers with a standard 12 hour on – 12 hour off cycle. |
|---|---|
| Wild animals | The study did not involve wild animals. |
| Reporting on sex | Male CD-1 mice were used for PK studies. Female CD-1 mice were used for MTD and thigh infection model. The mice are housed individually with free access to food and water during the study. Animals will be cared for in accordance with Guide for Care and Use of Laboratory Animals" (National Academy Press, Washington DC, 2011). |
| Field-collected samples | This study did not involved samples collected from the field. |
| Ethics oversight | Pharmaron Inc. carried out PK studies according to industry standards (Beijing, ROC). University of North Texas Health Science Center (Fort Worth, Texas) carried out infection model and MTD studies under UNTHSC approved (IACUC) protocols# IACUC-2021-0003 and IACUC-2020-0039. All procedures were conducted in accordance with a protocol approved by the UNTHSC Animal Care and Use Committee (IACUC) |

Note that full information on the approval of the study protocol must also be provided in the manuscript.

# Flow Cytometry

## Plots

Confirm that:

☒ The axis labels state the marker and fluorochrome used (e.g. CD4-FITC).

☒ The axis scales are clearly visible. Include numbers along axes only for bottom left plot of group (a 'group' is an analysis of identical markers).

☒ All plots are contour plots with outliers or pseudocolor plots.

☒ A numerical value for number of cells or percentage (with statistics) is provided.

## Methodology

| Sample preparation | Overnight P. aeruginosa PA14 or E. coli lptD4213 cultures were diluted 1:100 and grown to mid exponential phase at 37°C. Cultures were diluted 1:10 into PBS and treated with antibiotics for 15 minutes. P. aeruginosa PA14 was stained with TO-PRO-3 (640 nm excitation, 670/30 nm emission) to measure cell membrane integrity. E. coli lptD4213 was stained with both TO-PRO-3 and DiOC2(3) |
|---|---|
| Instrument | LSRII flow cytometer BD Biosciences |
| Software | FlowJo v10 software (FlowJo LLC, Ashland, OR) |
| Cell population abundance | For each assay 100,000 events were collected and cell population % is reported in each figure. |
| Gating strategy | Gates for depolarization were determined using CCCP as a positive control. Gates for permeabilization were determined using Polymixin B (Sigma-Aldrich P1004) and untreated controls. |

☒ Tick this box to confirm that a figure exemplifying the gating strategy is provided in the Supplementary Information.

