## [Peer Review File · Nature Microbiology]

Peer Review Information

Journal: Nature Microbiology

Manuscript Title: A folate inhibitor exploits metabolic differences in *Pseudomonas aeruginosa* for narrow-spectrum targeting

Corresponding author name(s): Dr Zemer Gitai

Editorial Notes:

This manuscript has been previously reviewed at another journal. This document only contains reviewer comments, rebuttal and decision letters for versions considered at Nature Microbiology. Mentions of the other journal have been redacted.

Reviewer Comments & Decisions:

Decision Letter, initial version:

Message: 15th November 2023

Dear Zemer,

Thank you for your patience while your manuscript "A novel inhibitor of *P. aeruginosa* folate metabolism exploits metabolic differences for narrow-spectrum antibiotic targeting" was under peer-review at Nature Microbiology. It has now been seen by 2 referees (that reviewed the previous submission at Nature), whose comments you will find at the of this email. You will see from their comments below that while they find your work of interest, some important points are raised. We are very interested in the possibility of publishing your study in Nature Microbiology, but would like to consider your response to these concerns in the form of a revised manuscript before we make a final decision on publication.

In particular, you will see that the referees ask to provide fluorofolin cytotoxicity data, to use fluorofolin instead of IRS143 (Fig 1I), to provide MIC of the mutants, to increase the limit of detection for some of the experiments, to determine synergy between fluorofilin + TMP experimentally, and to use another comparator than ciprofloxacin. Please note that editorially, we will require you to address these concerns experimentally. Please also temper the conclusions around resistance and provide a clear discussion of the limitations here. However, the referee also ask to determine the frequency of resistant strains form a random collection of clinical isolates and to use a mouse infection model to confirm the data. Please note, that editorially, we will not require you to address these points. The rest of the referees' reports are clear and the remaining issues should be straightforward to address.

If you have not done so already please begin to revise your manuscript so that it conforms to our Article format instructions at <http://www.nature.com/nmicrobiol/info/final-submission/>

The usual length limit for a Nature Microbiology Article is six display items (figures or tables) and 3,000 words. We have some flexibility, and can allow a revised manuscript at 3,500 words, but please consider this a firm upper limit. There is a trade-off of ~250 words per display item, so if you need more space, you could move a Figure or Table to Supplementary Information.

2Some reduction could be achieved by focusing any introductory material and moving it to the start of your opening 'bold' paragraph, whose function is to outline the background to your work, describe in a sentence your new observations, and explain your main conclusions. The discussion should also be limited. Methods should be described in a separate section following the discussion, we do not place a word limit on Methods.

Nature Microbiology titles should give a sense of the main new findings of a manuscript, and should not contain punctuation. Please keep in mind that we strongly discourage active verbs in titles, and that they should ideally fit within 90 characters each (including spaces).

Please include a data availability statement as a separate section after Methods but before references, under the heading "Data Availability". This section should inform readers about the availability of the data used to support the conclusions of your study. This information includes accession codes to public repositories (data banks for protein, DNA or RNA sequences, microarray, proteomics data etc...), references to source data published alongside the paper, unique identifiers such as URLs to data repository entries, or data set DOIs, and any other statement about data availability. At a minimum, you should include the following statement: "The data that support the findings of this study are available from the corresponding author upon request", mentioning any restrictions on availability. If DOIs are provided, we also strongly encourage including these in the Reference list (authors, title, publisher (repository name), identifier, year). For more guidance on how to write this section please see:

<http://www.nature.com/authors/policies/data/data-availability-statements-data-citations.pdf>

To improve the accessibility of your paper to readers from other research areas, please pay particular attention to the wording of the paper's opening bold paragraph, which serves both as an introduction and as a brief, non-technical summary in about 150 words. If, however, you require one or two extra sentences to explain your work clearly, please include them even if the paragraph is over-length as a result. The opening paragraph should not contain references. Because scientists from other sub-disciplines will be interested in your results and their implications, it is important to explain essential but specialised terms concisely. We suggest you show your summary paragraph to colleagues in other fields to uncover any problematic concepts.

If your paper is accepted for publication, we will edit your display items electronically so they conform to our house style and will reproduce clearly in print. If necessary, we will re-size figures to fit single or double column width. If your figures contain several parts, the parts should form a neat rectangle when assembled. Choosing the right electronic format at this

stage will speed up the processing of your paper and give the best possible results in print. We would like the figures to be supplied as vector files - EPS, PDF, AI or postscript (PS) file formats (not raster or bitmap files), preferably generated with vector-graphics software (Adobe Illustrator for example). Please try to ensure that all figures are non-flattened and fully editable. All images should be at least 300 dpi resolution (when figures are scaled to approximately the size that they are to be printed at) and in RGB colour format. Please do not submit Jpeg or flattened TIFF files. Please see also 'Guidelines for Electronic Submission of Figures' at the end of this letter for further detail.

Figure legends must provide a brief description of the figure and the symbols used, within 350 words, including definitions of any error bars employed in the figures.

When submitting the revised version of your manuscript, please pay close attention to our [href="https://www.nature.com/nature-research/editorial-policies/image-integrity">Digital Image Integrity Guidelines](https://www.nature.com/nature-research/editorial-policies/image-integrity). and to the following points below:

Please include a statement before the acknowledgements naming the author to whom correspondence and requests for materials should be addressed.

Finally, we require authors to include a statement of their individual contributions to the paper -- such as experimental work, project planning, data analysis, etc. -- immediately after the acknowledgements. The statement should be short, and refer to authors by their initials. For details please see the Authorship section of our joint Editorial policies at http://www.nature.com/authors/editorial_policies/authorship.html

- * include a point-by-point response to any editorial suggestions and to our referees. Please include your response to the editorial suggestions in your cover letter, and please upload your response to the referees as a separate document.
- * ensure it complies with our format requirements for Letters as set out in our guide to authors at www.nature.com/nmicrobiol/info/gta/
- * state in a cover note the length of the text, methods and legends; the number of references; number and estimated final size of figures and tables

* resubmit electronically if possible using the link below to access your home page:

*This url links to your confidential homepage and associated information about manuscripts you may have submitted or be reviewing for us. If you wish to forward this e-mail to co-authors, please delete this link to your homepage first.

Please ensure that all correspondence is marked with your Nature Microbiology reference number in the subject line.

Nature Microbiology is committed to improving transparency in authorship. As part of our efforts in this direction, we are now requesting that all authors identified as 'corresponding author' on published papers create and link their Open Researcher and Contributor Identifier (ORCID) with their account on the Manuscript Tracking System (MTS), prior to acceptance. This applies to primary research papers only. ORCID helps the scientific community achieve unambiguous attribution of all scholarly contributions. You can create and link your ORCID from the home page of the MTS by clicking on 'Modify my Springer Nature account'. For more information please visit www.springernature.com/orcid.

We hope to receive your revised paper within three months. If you cannot send it within this time, please let us know.

Reviewers Comments:

Reviewer #1 (Remarks to the Author):

This is an improved manuscript describing fluorofolin, a novel inhibitor of dihydrofolate reductase in *P. aeruginosa*. The authors suggest that this inhibitor, while not specific to *P. aeruginosa* DHFR, may selectively inhibit growth of this pathogen, since it can not use external sources of thymidine. This is a clever argument, but it is somewhat undermined by the high frequency of mutations conferring resistance to fluorofolin, which will necessitate a combination with broad-spectrum sulfamethoxazole. Specific comments follow.

1. Cytotoxicity of fluorofolin is examined against several human cell lines, but the data are not presented. Instead, the ratio of fluorofolin cytotoxicity vs toxicity of another experimental compound is given (Suppl 2D); the relevance of this is unclear. Please provide fluorofolin cytotoxicity.

2. "However, these mutants did not decrease their MIC against IRS143 to the same extent (Figure 1I),"

5- Please use fluorofolin instead of IRS143.

3. "We also note TMP still has a higher MIC than fluorofolin in mexA, mexB and oprM mutants, suggesting that fluorofolin may also improve other features of cell accumulation such as membrane penetrance or interactions with other efflux pumps"

- This is unclear. The simple possibility is that fluorofolin, unlike TMP, is not a good substrate of numerous MDR pumps present in *P. aeruginosa*.

4. A key concern regarding fluorofolin is the very high frequency of mutations due to overexpression of MDR pumps. The authors address this by showing that virulence of such mutants is diminished, and that a combination of fluorofolin + TMP results in low frequency of resistance. They additionally state that clinical isolates of *P. aeruginosa* have a low probability of being resistant to fluorofolin. The manuscript will benefit from substantiating and documenting these findings. Specifically:

a. The authors report selecting for mutants resistant to fluorofolin, but surprisingly do not present essential data, MIC of these mutants.

b. A combination of fluorofolin + TMP does not produce resistant mutants, with a limit of detection 10⁻⁸. This is not a reasonable measure, typically, a culture is concentrated and plated at 10⁹ cells/plate, and preparing 10 plates then gives a 10⁻¹⁰ limit of detection. Synergy between fluorofolin + TMP is indeed likely to produce low levels of resistant mutants, but this is important to properly determine experimentally.

c. 14.5% of *P. aeruginosa* clinical isolates were resistant to fluorofolin due to the presence of a mutant allele of DHFR. The authors note that such mutants are rare. While a 14% failure rate is acceptable for an old antibiotic, this is not the case for a novel compound. At the same time, these clinical isolates are not random, but taken from the antibiotic resistance isolate bank. It is important to determine the frequency of resistant strains from a random collection of clinical isolates; it is quite likely that in this case, more relevant to evaluating the potential of a compound, the frequency of resistant isolate will be much lower. In determining resistance, please report MIC₉₀, a standard measure in such evaluations which reports the minimal concentration of compound needed to inhibit ≥90% of isolates.

d. Not clear how virulence in a *C. elegans* model translates into mammalian virulence. Please perform this test with mice.

e. The authors argue that clinical isolates of *P. aeruginosa* with overexpression of MDR pumps are rare, suggesting diminished fitness, and provide a reference. This argument is not convincing, there are many other studies showing that *P. aeruginosa* with upregulated MDR pumps are common, for example, resistance to meropenem requires both a mutation in the OprD porin and an upregulated MDR pump.

Reviewer #2 (Remarks to the Author):

Fluorofolin is a novel dihydrofolate reductase inhibitor that shows narrow spectrum inhibition of *Pseudomonas* when administered in the presence of thymine. The previous reviewers felt that the lead compound, now called fluorofolin, was not qualified as a clinical candidate. The authors have made a very reasonable rebuttal that such qualification has not generally been required for the initial publication of a compound with novel properties that may introduce a new therapeutic concept – in this case applying metabolic information to create a narrow spectrum agent – and this one in *Pseudomonas*. The finding that fluorofolin will inhibit *Pseudomonas* but not *E. coli* or other enterics or Staph, if dosed in the presence of thymine/methionine/inosine [since *Pseudomonas* lacks the enzymes to incorporate exogenous thymine] very interesting and it appears that this finding may be successful in developing a *Pseudomonas*-specific drug. Thus, I am not negative about the lack of some aspects, but I still feel there is some further worry about resistance. First, the selection for single step resistance was done using 10^8 bacteria, so mutations occurring at a FoR of 1×10^{-9} would not be seen. Would mutations arise in *Pseudomonas* DHFR itself? Second, ciprofloxacin is not a reasonable comparator. It inhibits both gyrase and topoisomerase IV; its low MIC is due to its inhibition of gyrase, but gyrase mutations alone do not normally raise the MIC very much; further mutations -in efflux and topoisomerase IV – are required to give clinically relevant resistance. The very high FoRs for fosfofolin and ciprofloxacin seen at 4x MIC are surprisingly, even for in vitro measurements. Third, the finding that (line 274 and ff) 14.5% of clinical isolates are already resistant to flourofolin is not a good sign. While it is noted that cipro resistance is 21.9%, that drug has been in use for >25 years and such a resistance level would preclude its use in empirical treatment. The lack of detection of efflux mutants in the clinical isolates does seem to agree with their probable lowered virulence. But it is the presence of the *dfrB5* cassette with a mutant DHFR allele (line 279) that is already present among *Pseudomonas* and the possibility of selection by fluorofolin of mutated *Pseudomonas* with resistant DHFR that are more worrying for development of fluorofolin. Nevertheless, as noted above, it is not necessary that the compound be perfect to be publishable – however, the likelihood that DHFR mutations will eventually overtake fluorofolin should be mentioned. Sulfamethoxazole may preserve activity, however.

Author Rebuttal to Initial comments

General comments to reviewers:

We greatly appreciate the constructive feedback of both reviewers. Below we address each of the

7reviewers' comments individually.

Reviewer #1 (Remarks to the Author):

This is an improved manuscript describing fluorofolin, a novel inhibitor of dihydrofolate reductase in *P. aeruginosa*. The authors suggest that this inhibitor, while not specific to *P. aeruginosa* DHFR, may selectively inhibit growth of this pathogen, since it cannot use external sources of thymidine. This is a clever argument, but it is somewhat undermined by the high frequency of mutations conferring resistance to fluorofolin, which will necessitate a combination with broad-spectrum sulfamethoxazole. Specific comments follow.

1. Cytotoxicity of fluorofolin is examined against several human cell lines, but the data are not presented. Instead, the ratio of fluorofolin cytotoxicity vs toxicity of another experimental compound is given (Suppl 2D); the relevance of this is unclear. Please provide fluorofolin cytotoxicity.

As suggested, we have now added the fluorofolin human cell line toxicity data to Supplemental Figure 2D. The ratio in cytotoxicity to IRS-16 is also provided to demonstrate a reduction in cytotoxicity compared to previous derivatives of fluorofolin, as fluorofolin no longer disrupts the cell membrane. As DHFR is also a common target in cancer therapeutics, it is not surprising that growth of some *in vitro* cell lines can also be inhibited by fluorofolin to some extent.

2. "However, these mutants did not decrease their MIC against IRS143 to the same extent (Figure 1I),"

- Please use fluorofolin instead of IRS143.

We thank the reviewer for catching this mistake and have edited the text accordingly.

3. "We also note TMP still has a higher MIC than fluorofolin in *mexA*, *mexB* and *oprM* mutants, suggesting that fluorofolin may also improve other features of cell accumulation such as membrane penetrance or interactions with other efflux pumps"

- This is unclear. The simple possibility is that fluorofolin, unlike TMP, is not a good substrate of numerous MDR pumps present in *P. aeruginosa*.

It is certainly possible that fluorofolin is not as good of a substrate for other efflux pumps that were not directly tested here. It is also possible that fluorofolin is much better than TMP at penetrating the cell envelope of *P. aeruginosa*. However, untangling these two possibilities is difficult to test experimentally so the text has been edited to clarify both possibilities.

4. A key concern regarding fluorofolin is the very high frequency of mutations due to overexpression of MDR pumps. The authors address this by showing that virulence of suchmutants is diminished, and that a combination of fluorofolin + TMP results in low frequency of resistance. They additionally state that clinical isolates of *P. aeruginosa* have a low probability of being resistant to fluorofolin. The manuscript will benefit from substantiating and documenting these findings. Specifically:

a. The authors report selecting for mutants resistant to fluorofolin, but surprisingly do not present essential data, MIC of these mutants.

The MIC of the *nfxB* mutant is greater than the highest concentration tested (>100 µg/mL) and the MIC of the *mexS* mutant is 62.5µg/mL. These results have been incorporated into the text.

b. A combination of fluorofolin + TMP does not produce resistant mutants, with a limit of detection 10⁻⁸. This is not a reasonable measure, typically, a culture is concentrated and plated at 10⁹ cells/plate, and preparing 10 plates then gives a 10⁻¹⁰ limit of detection. Synergy between fluorofolin + TMP is indeed likely to produce low levels of resistant mutants, but this is important to properly determine experimentally.

As requested, we have repeated the resistance plating of the 4X MIC fluorofolin/SMX combination on ten plates containing 10⁹ CFU per plate. We determined that the resistance frequency using these conditions was <1 in 10¹⁰ as no colonies were isolated. These new results have been incorporated into the revised text.

c. 14.5% of *P. aeruginosa* clinical isolates were resistant to fluorofolin due to the presence of a mutant allele of DHFR. The authors note that such mutants are rare. While a 14% failure rate is acceptable for an old antibiotic, this is not the case for a novel compound. At the same time, these clinical isolates are not random, but taken from the antibiotic resistance isolate bank. It is important to determine the frequency of resistant strains from a random collection of clinical isolates; it is quite likely that in this case, more relevant to evaluating the potential of a compound, the frequency of resistant isolate will be much lower. In determining resistance, please report MIC₉₀, a standard measure in such evaluations which reports the minimal concentration of compound needed to inhibit ≥90% of isolates.

We thank both the reviewer for raising this interesting question and the editor for noting that this effort lies outside the scope of the current work. Nevertheless, we were able to access a collection of 22 clinical *P. aeruginosa* isolates from blood. Encouragingly, fluorofolin successfully inhibited the growth of all 22, supporting our conclusion that there is not widespread resistance already present in clinical isolates. This result also further supports our hypothesis that *mexCD-oprN* and *mexEF-oprJ* mutants are rare in clinical isolates,

as these strains were all sensitive to ciprofloxacin and fluorofolin. Including the new 22 clinical isolates, fluorofolin inhibits 69 out of the 77 within the entire collection (90%) at 50µg/mL (Figure 4F). We did not test our collection against lower concentrations of fluorofolin, so we can claim that the MIC₉₀ is less

10than or equal to 50µg/mL. Because the MIC90 value is greatlydependent on the library composition, we have not included the MIC90 in the revised text as we do not believe it significantly adds to the paper.

d. Not clear how virulence in a *C. elegans* model translates into mammalian virulence. Please perform this test with mice.

We thank the editor for noting that we do not need to repeat this experiment in a mouse model and have added references indicating that the *C. elegans* model translates into mammalian virulence (PMID: 10051655).

e. The authors argue that clinical isolates of *P. aeruginosa* with overexpression of MDR pumps are rare, suggesting diminished fitness, and provide a reference. This argument is not convincing, there are many other studies showing that *P. aeruginosa* with upregulated MDR pumps are common, for example, resistance to meropenem requires both a mutation in the OprD porin and an upregulated MDR pump.

We note that in our study we examined 77 *P. aeruginosa* clinical isolates (including the additional 22 added in the revision) and found no evidence that any of them are efflux pump overexpressors (none are resistant to both ciprofloxacin and fluorofolin). This is consistent with other studies, such as PMID: 18474583, which found that only 4/110 of the ciprofloxacin resistant clinical isolates tested had increased expression of MexCD-OprJ expression due to *nfxB* mutation. Nevertheless, we see the reviewer's point that there are other examples, especially in chronic infections like CF, where there are reports of elevated efflux pump expression (for example PMID: 10681343).

Consequently, we have revised the text to acknowledge the likelihood that in certain niches efflux pump overexpressors are able to survive, but due to many fitness and virulence costs, they may be rare depending on which types of infections the clinical isolates are derived from. We have also cited additional studies on the growing interest in the field of exploiting collateral sensitivity (CS) of mutant strains in which resistance to one antibiotic confers sensitivity to another. Efflux pump overexpressing strains of *P. aeruginosa* with mutations in *nfxB* have been shown to have CS towards β -lactam and aminoglycoside antibiotics (PMID: 36997518, PMID: 29307490) suggesting that antibiotic cycling may also be able to overcome the limitations of fluorofolin resistance.

Reviewer #2 (Remarks to the Author):

Fluorofolin is a novel dihydrofolate reductase inhibitor that shows narrow spectrum inhibition of *Pseudomonas* when administered in the presence of thymine.

The previous reviewers felt that the lead compound, now called fluorofolin, was not qualified as a

12clinical candidate. The authors have made a very reasonable rebuttal that such qualification has not generally been required for the initial publication of a compound with novel properties that may introduce a new therapeutic concept – in this case applying metabolic information to create a narrow spectrum agent – and this one in *Pseudomonas*. The finding that fluorofolin will inhibit *Pseudomonas* but not *E. coli* or other enterics or Staph, if dosed in the presence ofthymine/methionine/inosine [since *Pseudomonas* lacks the enzymes to incorporate exogenous thymine] very interesting and it appears that this finding may be successful in developing a *Pseudomonas*-specific drug. Thus, I am not negative about the lack of some aspects, but I still feel there is some further worry about resistance.

First, the selection for single step resistance was done using 10^8 bacteria, so mutations occurring at a FoR of 1×10^{-9} would not be seen. Would mutations arise in *Pseudomonas* DHFR itself? Second, ciprofloxacin is not a reasonable comparator. It inhibits both gyrase and topoisomerase IV; its low MIC is due to its inhibition of gyrase, but gyrase mutations alone do not normally raise the MIC very much; further mutations -in efflux and topoisomerase IV – are required to give clinically relevant resistance.

As requested, we have repeated the resistance plating of the 4X MIC fluorofolin/SMX combination on ten plates containing 10^9 CFU per plate. We determined that the resistance frequency using these conditions was <1 in 10^{10} as no colonies were isolated. We were not able to detect DHFR mutations in these conditions, likely because the frequency of efflux pump upregulation masked any other types of mutations.

We also repeated our resistance plating using gentamycin and meropenem to determine resistance frequencies of two other classes of antibiotics that have single targets as better comparators to fluorofolin. We found that fluorofolin had a significantly lower resistance frequency than ciprofloxacin and meropenem and a comparable resistance frequency to gentamycin (Figure 4C). We also tested both gentamycin and meropenem against the clinical isolate panel to calculate resistance frequencies to these antibiotics (Figure 4G). We again found that fluorofolin performed favorably in comparison to these antibiotics.

The very high FoRs for fosfofolin and ciprofloxacin seen at 4x MIC are surprisingly, even for in vitro measurements. Third, the finding that (line 274 and ff) 14.5% of clinical isolates are already resistant to flourofolin is not a good sign. While it is noted that cipro resistance is 21.9%, that drug has been in use for >25 years and such a resistance level would preclude its use in empirical treatment. The lack of detection of efflux mutants in the clinical isolates does seem to agree with their probable lowered virulence. But it is the presence of the *dfrB5* cassette with a mutant DHFR allele (line 279) that is already present among *Pseudomonas* and the possibility of selection by fluorofolin of mutated *Pseudomonas* with resistant DHFR that are more worrying for development of fluorofolin. Nevertheless, as noted above, it is not necessary that the compound be perfect to be publishable – however, the likelihood that DHFR mutations will eventually overtake fluorofolin should be mentioned. Sulfamethoxazole may preserve activity, however.

We thank the reviewer for this point. First, as noted by Reviewer #1, it is important to remember that the CDC panel is specifically enriched for MDR isolates. We have now tested an additional 22 randomly-

collected clinical isolates and found that none of them were resistant to fluorofolin (see response to comment 4C of Reviewer 1 for more details). Furthermore, we agree with the reviewer that hopefully the cotreatment with SMX and good antibiotic stewardship could at least slow down the rate at which DHFR mutations would emerge. Nevertheless, we understand the reviewer's point and have thus discussed the concern surrounding existing *dfrB5* mutations more explicitly in the text. Finally, as also suggested by Reviewer #1, we have also added an additional section in the discussion to temper some of the conclusions regarding fluorofolin resistance.

Decision Letter, first revision:

Message: Our ref: NMICROBIOL-23102763A

6th February 2024

Dear Zemer,

Thank you for your patience as we've prepared the guidelines for final submission of your Nature Microbiology manuscript, "A novel *P. aeruginosa* folate inhibitor exploits metabolic differences for narrow-spectrum targeting" (NMICROBIOL-23102763A). Please carefully follow the step-by-step instructions provided in the attached file, and add a response in each row of the table to indicate the changes that you have made. Ensuring that each point is addressed will help to ensure that your revised manuscript can be swiftly handed over to our production team.

In recognition of the time and expertise our reviewers provide to Nature Microbiology's editorial process, we would like to formally acknowledge their contribution to the external peer review of your manuscript entitled "A novel *P. aeruginosa* folate inhibitor exploits metabolic differences for narrow-spectrum targeting". For those reviewers who give their assent, we will be publishing their names alongside the published article.

15Nature Microbiology offers a Transparent Peer Review option for new original research manuscripts submitted after December 1st, 2019. As part of this initiative, we encourage our authors to support increased transparency into the peer review process by agreeing to have the reviewer comments, author rebuttal letters, and editorial decision letters published as a Supplementary item. When you submit your final files please clearly state in your cover letter whether or not you would like to participate in this initiative. Please note that failure to state your preference will result in delays in accepting your manuscript for publication.

Cover suggestions

COVER ARTWORK: We welcome submissions of artwork for consideration for our cover. For more information, please see our guide for cover artwork.

Nature Microbiology has now transitioned to a unified Rights Collection system which will allow our Author Services team to quickly and easily collect the rights and permissions required to publish your work. Approximately 10 days after your paper is formally accepted, you will receive an email in providing you with a link to complete the grant of rights. If your paper is eligible for Open Access, our Author Services team will also be in touch regarding any additional information that may be required to arrange payment for your article.

Please note that *Nature Microbiology* is a Transformative Journal (TJ). Authors may publish their research with us through the traditional subscription access route or make their paper immediately open access through payment of an article-processing charge (APC). Authors will not be required to make a final decision about access to their article until it has been accepted. Find out more about Transformative Journals

Reviewer #1:

Remarks to the Author:

The authors have adequately responded to critiques.

Reviewer #2:

Remarks to the Author:

This revised paper concerns the compound fluorofolin which is shown to be an inhibitor of dihydrofolate reductase (DHFR). Fluorofolin was based on the previously discovered IRS-16 which had been shown to inhibit both DHFR and to have membrane lytic activities. These multiple activities endowed IRS-16 with very low resistance potential, but the membrane activity led to toxicity/tolerability problems. With fluorofolin having been optimized to have only one main target, the possibility of rapid resistance development is a theoretical worry (see below). The authors emphasize that fluorofolin can have a narrow spectrum of activity against *P. aeruginosa* if given in the presence of thymine, since *P. aeruginosa* lacks the genes necessary to scavenge thymine while most other species can do so.

The previous reviewers mentioned the importance of assessing resistance potential. The authors did try to address this. In vitro, mutations leading to overexpression of various efflux pumps caused fluorofolin (and ciprofloxacin) resistance at relatively high frequency of resistance (FoR). It was found that such mutations also lead to reduced growth rates and lowered virulence. The authors propose that such mutants would be unlikely to arise in the clinic. They cite the rate of *P. aeruginosa* resistance to various drugs as much higher than the 10.4% resistance to fluorofolin (at 50 mg/liter). But these other drugs have been in use for decades – and it could be argued that 10.4% is too high for a newly introduced drug. It seems likely that further rounds of selection with fluorofolin could lead to compensatory mutations that restore growth and virulence to the efflux overexpressers.

It is interesting that the authors found no DHFR mutations in their resistance selection study (presumably due to the high level of efflux regulatory mutations). On the other hand, among the preexisting resistant *P. aeruginosa* (the 10.4%), they found a preponderance of isolates bearing a transposon carrying a resistant DHFR gene. What is the likelihood that such clones would not be selected upon clinical use of fluorofolin? Recognizing that there is a likelihood of resistance development, the authors tested synergy with sulfamethoxazole to increase the efficacy of fluorofolin in the animal model and propose that the use of sulfamethoxazole would reduce the in vivo FoR (as demonstrated in vitro). I basically agree with the plans to use combinations of drugs to prevent or reduce resistance (as is done with MTB, HIV, HCV, cancer). However addition of sulfamethoxazole to synergize fluorofolin obviates the proposed benefits of narrow spectrum (which should be discussed).

It is unlikely that fluorofolin itself could be a drug candidate, but the concept of narrowing spectrum based on specific metabolic patterns is worth publishing. It might be possible to reduce the effluxability of fluorofolin or even a DHFR inhibitor of a different scaffold to reduce the selection of efflux upregulation – and this should be mentioned.

Author Rebuttal, first revision:

General comment to reviewers:

We greatly appreciate the constructive feedback of both reviewers. Below we address each of the reviewers' comments individually.

Point-by-point reply to reviewers:

Reviewer #1 (Remarks to the Author):

The authors have adequately responded to critiques.

We thank the reviewer for their feedback throughout the editorial process.

Reviewer #2 (Remarks to the Author):

This revised paper concerns the compound fluorofolin which is shown to be an inhibitor of dihydrofolate reductase (DHFR). Fluorofolin was based on the previously discovered IRS-16 which had been shown to inhibit both DHFR and to have membrane lytic activities. These multiple activities endowed IRS-16 with very low resistance potential, but the membrane activity led to toxicity/tolerability problems. With fluorofolin having been optimized to have only one main target, the possibility of rapid resistance development is a theoretical worry (see below). The authors emphasize that fluorofolin can have a narrow spectrum of activity against *P. aeruginosa* if given in the presence of thymine, since *P. aeruginosa* lacks the genes necessary to scavenge thymine while most other species can do so.

We thank the reviewer for this summary of the edited manuscript.

The previous reviewers mentioned the importance of assessing resistance potential. The authors did try to address this. *In vitro*, mutations leading to overexpression of various efflux pumps caused fluorofolin (and ciprofloxacin) resistance at relatively high frequency of resistance (FoR). It was found that such mutations also lead to reduced growth rates and lowered virulence. The authors propose that such mutants would be unlikely to arise in the clinic. They cite the rate of *P. aeruginosa* resistance to various drugs as much higher than the 10.4% resistance to fluorofolin (at 50 mg/liter). But these other drugs have been in use for decades – and it could be argued that 10.4% is too high for a newly introduced drug. It seems likely that

further rounds of selection with fluorofolin could lead to compensatory mutations that restore growth and virulence to the efflux overexpressers. It is interesting that the authors found no DHFR mutations in their resistance selection study (presumably due to the high level of efflux regulatory mutations). On the other hand, among the preexisting resistant *P. aeruginosa* (the 10.4%), they found a preponderance of isolates bearing a transposon carrying a resistant DHFR gene. What is the likelihood that such clones would not be selected upon clinical use of fluorofolin?

We agree that 10.4% of clinical strains showing resistance is cause for concern. However, all of the strains that were resistant to fluorofolin were amongst the CDC collection, which has been curated by the CDC to contain multi-drug resistant strains. This number likely overestimates what one would expect to see in the clinic as all the non-multidrug resistant blood isolates tested were sensitive to fluorofolin. We mention in the discussion that long term clinical use of fluorofolin could select for increased prevalence of the *dfrB5* allele, but cotreatment with SMX could help to mitigate this issue.

Recognizing that there is a likelihood of resistance development, the authors tested synergy with sulfamethoxazole to increase the efficacy of fluorofolin in the animal model and propose that the use of sulfamethoxazole would reduce the in vivo FoR (as demonstrated in vitro). I basically agree with the plans to use combinations of drugs to prevent or reduce resistance (as is done with MTB, HIV, HCV, cancer). However addition of sulfamethoxazole to synergize fluorofolin obviates the proposed benefits of narrow spectrum (which should be discussed).

Addition of sulfamethoxazole should not reduce the proposed benefits of narrow spectrum fluorofolin. In *E. coli*, both thymine and thymidine can rescue from treatment with the combination of fluorofolin and SMX (Extended Data Figure 3G). We expect this should be true of other bacterial species able to utilize thymine as SMX inhibits dihydropteroate synthetase, which lies directly upstream of DHFR in the tetrahydrofolate metabolic pathway.

It is unlikely that fluorofolin itself could be a drug candidate, but the concept of narrowing spectrum based on specific metabolic patterns is worth publishing. It might be possible to reduce the effluxability of fluorofolin or even a DHFR inhibitor of a different scaffold to reduce the selection of efflux upregulation – and this should be mentioned.

We have added a sentence in the discussion to comment on the possibility of further scaffold modification to select for derivatives that are less likely to be effluxed.

Final Decision Letter:

Message: 6th March 2024

Dear Zemer,

I am pleased to accept your Article "A folate inhibitor exploits metabolic differences in *Pseudomonas aeruginosa* for narrow-spectrum targeting" for publication in *Nature Microbiology*. Thank you for having chosen to submit your work to us and many congratulations.

Over the next few weeks, your paper will be copyedited to ensure that it conforms to *Nature Microbiology* style. We look particularly carefully at the titles of all papers to ensure that they are relatively brief and understandable.

Please note that *Nature Microbiology* is a Transformative Journal (TJ). Authors may publish their research with us through the traditional subscription access route or make their paper immediately open access through payment of an article-processing charge (APC). Authors

2will not be required to make a final decision about access to their article until it has been accepted. Find out more about Transformative Journals

Congratulations once again and I look forward to seeing the article published.

With kind regards,